# TMK-based cell-surface auxin signalling activates cell-wall acidification

Wenwei Lin[1,2], Xiang Zhou[1,2], Wenxin Tang[1], Koji Takahashi[3,4], Xue Pan[2], Jiawei Dai[1], Hong Ren[5], Xiaoyue Zhu[1], Songqin Pan[2], Haiyan Zheng[6], William M. Gray[5], Tongda Xu[1], Toshinori Kinoshita[3,4] & Zhenbiao Yang[1,2]✉

The phytohormone auxin controls many processes in plants, at least in part through its regulation of cell expansion[1]. The acid growth hypothesis has been proposed to explain auxin-stimulated cell expansion for five decades, but the mechanism that underlies auxin-induced cell-wall acidification is poorly characterized. Auxin induces the phosphorylation and activation of the plasma membrane H⁺-ATPase that pumps protons into the apoplast[2], yet how auxin activates its phosphorylation remains unclear. Here we show that the transmembrane kinase (TMK) auxin-signalling proteins interact with plasma membrane H⁺-ATPases, inducing their phosphorylation, and thereby promoting cell-wall acidification and hypocotyl cell elongation in *Arabidopsis*. Auxin induced interactions between TMKs and H⁺-ATPases in the plasma membrane within seconds, as well as TMK-dependent phosphorylation of the penultimate threonine residue on the H+-ATPases. Our genetic, biochemical and molecular evidence demonstrates that TMKs directly phosphorylate plasma membrane H⁺-ATPase and are required for auxin-induced H⁺-ATPase activation, apoplastic acidification and cell expansion. Thus, our findings reveal a crucial connection between auxin and plasma membrane H⁺-ATPase activation in regulating apoplastic pH changes and cell expansion through TMK-based cell surface auxin signalling.

Embedded in a rigid cell wall, the plant cell must modify its wall to gain the adjustable elasticity to regulate cell expansion in space and time. Auxin induces rapid cell expansion by acidifying the cell-wall space (apoplast), leading to the activation of cell-wall-localized proteins for wall loosening[3,4], a growth mechanism that has been known as the acid growth theory for half of a century[5]. Auxin triggers the efflux of protons, resulting in apoplastic acidification by activating the plasma membrane (PM)-localized P-type H⁺-ATPase[6,7]. In *Arabidopsis*, PM H⁺-ATPase is encoded by an autoinhibited H⁺-ATPase (AHA) gene family comprising 11 members[8]. Phosphorylation of the conserved penultimate Thr residue (Thr948 in AHA1, Thr947 in AHA2) has been proposed to release the autoinhibition of the ATPase pump activity by the cytoplasmic C-terminal region[9–15]. Fendrych et al. previously demonstrated that auxin-induced apoplastic acidification and growth are mediated by TIR1/AFB–Aux/IAA nuclear auxin perception in hypocotyls[16]. Auxin induces the TIR1/AFB-dependent expression of SAUR proteins that act as inhibitors of PP2C.D phosphatases, which dephosphorylate the penultimate Thr residue[17]. Although this mechanism can sustain H⁺-ATPase activity by preventing the dephosphorylation of the penultimate Thr residue, it cannot account for how the PM H⁺-ATPase is initially phosphorylated to become activated.

The PM-localized TMK-receptor-like kinases have a vital role in auxin signalling in regulating pavement cell morphogenesis, differential growth of the apical hook, lateral root formation and root thermomorphogenesis in *Arabidopsis*[18–23]. Auxin rapidly promotes TMK-dependent activation of PM-associated ROP GTPases within seconds, providing a mechanism for rapid auxin responses on the cell surface in addition to TIR1/AFB-based intracellular auxin signalling[20,24–26]. To identify new components in TMK-mediated auxin-signalling pathways, we performed immunoprecipitation coupled with mass spectrometry (IP–MS) to isolate potential interactors of TMK1 in *Arabidopsis*. In brief, GFP-Trap agarose beads were used to immunoprecipitate the TMK1–GFP protein complex from *pTMK1::TMK1-GFP* transgenic plants, which was further analysed using MS. The proteins that were identified only from the *pTMK1::TMK1-GFP* transgenic plants but not from the *pTMK1::GFP* control plants were considered to be candidates for TMK1-associated proteins (Supplementary Table 2). Among them, we were especially interested in the PM H⁺-ATPases (AHAs) (Extended Data Fig. 1a), as the previous study showed that auxin triggers the activation of the PM H⁺-ATPase, which promotes hypocotyl cell elongation[27]. We further confirmed that GFP–AHA1 co-immunoprecipitated with TMK1 and TMK4 in the *35S::GFP-AHA1* transgenic plants as detected by immunoblotting using anti-TMK1 and anti-TMK4 antibodies, respectively (Fig. 1a and Extended Data Fig. 1b). Furthermore, TMK1–GFP co-immunoprecipitated with AHA(s) from *pTMK1::TMK1-GFP* transgenic plants as detected by immunoblot analysis using anti-AHA2-cat

[1]FAFU-UCR Joint Center for Horticultural Biology and Metabolomics, Haixia Institute of Science and Technology, Fujian Agriculture and Forestry University, Fuzhou, China. [2]Institute of Integrative Genome Biology and Department of Botany and Plant Science, University of California, Riverside, CA, USA. [3]Graduate School of Science, Nagoya University, Nagoya, Japan. [4]Institute of Transformative Bio-Molecules, Nagoya University, Nagoya, Japan. [5]Department of Plant and Microbial Biology, University of Minnesota, St Paul, MN, USA. [6]Biological Mass Spectrometry Facility, Robert Wood Johnson Medical School and Rutgers, the State University of New Jersey, Piscataway, NJ, USA. ✉e-mail: yang@ucr.edu

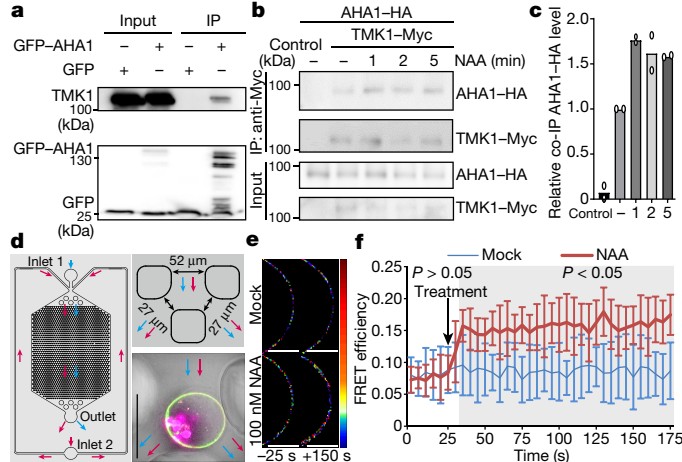

**Fig. 1 | TMK1 interacts directly with AHAs. a**, Co-IP analysis of TMK1 with GFP–AHA1. *3SS::GFP* (control) and *3SS::GFP-AHA1* plants were immunoprecipitated with anti-GFP antibodies and analysed by western blotting using anti-TMK1 antibodies. **b**, Auxin induced interactions between TMK1–Myc and AHA–HA in *Arabidopsis* protoplasts. AHA1–HA and TMK1–Myc constructs were transiently expressed in protoplasts, which were then treated with 1 μM NAA for 1, 2 and 5 min before being used for co-IP analysis. **c**, Quantification of AHA1–HA proteins co-immunoprecipitated with TMK1–Myc as shown in **b**. Data are the mean values of two independent biological replicates. **d**, The microfluidics device that was designed to investigate the auxin-induced rapid TMK1–AHA1 interaction using FRET analysis. Right, a triangle trap (top) for trapping a protoplast (bottom). The blue arrows indicate the flow of cell suspension, and the red arrows indicate the flow of NAA or mock solutions. Scale bar, 50 μm. **e**, FRET analysis of the rapid induction of the TMK1–AHA interaction. A representative heat map of sensitized emission efficiencies for the TMK1–mCherry/AHA1–GFP FRET on the PM region (further details are provided in Extended Data Fig. 1d). The times before and after treatment are indicated (−25 s and +150 s). Scale bars, 10 μm. **f**, Quantitative time-course analyses of changes in the FRET efficiencies. NAA (100 nM) or mock buffer was applied at 25 s after imaging started. The error bars indicate the s.d. of 10 cells scored. Statistical analysis was performed using two-sided Student's *t*-tests (*P* < 0.05); the grey background indicates significant differences during the covered periods..

antibodies after anti-GFP-Trap antibody immunoprecipitation (Extended Data Fig. 1c). An in vitro pull-down assay showed that the kinase domain of TMK1 (TMK1[KD]), when fused to maltose-binding protein (MBP), directly interacted with the AHA2 C-terminal domain fused to glutathione *S*-transferase (GST) (GST–AHA2-C) (Extended Data Fig. 1d), suggesting that the kinase domain of TMK1 directly binds to the C-terminal region of AHA2. We postulated that TMK1 interacts with AHAs in vivo as both proteins are predominantly localized at the PM[20,28,29]. A co-immunoprecipitation (co-IP) assay showed that the in vivo association between Myc-tagged TMK1 and HA-tagged AHA1 was enhanced within 1 min after treatment with 1-naphthaleneacetic acid (NAA) (Fig. 1b, c), suggesting that auxin rapidly promotes TMK1–AHA interactions. To test whether auxin rapidly induces a direct interaction between TMK1 and AHA1, we developed a microfluidics device enabling the high-resolution time-course analysis of a rapid induction of the dynamic interaction in a single *Arabidopsis* protoplast using fluorescence resonance energy transfer (FRET) imaging (Fig. 1d). Protoplasts co-expressing TMK1–mCherry and AHA1–GFP were captured in triangle traps within the device, enabling imaging before and immediately after auxin treatment (Fig. 1d and Extended Data Fig. 1e). Intriguingly, the TMK1–mCherry/AHA1–GFP FRET efficiency increased within 10 s after auxin treatment, indicating that auxin very rapidly promotes the direct interaction between TMK1–mCherry and AHA1–GFP (Fig. 1e, f). By contrast, no increase in FRET efficiency was detected when protoplasts co-expressing TMK1–mCherry and AHA1–GFP were mock-treated with

control buffer (Fig. 1e, f). Together, these results show that auxin promotes a rapid and direct interaction between TMK1 and AHA1 on the PM.

The phosphorylation of the conserved penultimate threonine residue on the H⁺-ATPase proteins is a primary mechanism by which the H⁺-ATPase is activated in response to multiple signals, including phytohormones, sucrose, NaCl, blue light and the fungal toxin fusicoccin[13,14,27,30–32]. We first examined the phosphorylation status of the penultimate threonine residue in the aerial parts of *Arabidopsis* seedlings using phosphoproteomics (with the roots removed when the seedlings were prepared for the assay). The phosphorylation levels of the penultimate Thr residue of AHA2, AHA3 and AHA7 were compromised in the *tmk1-1 tmk4-1* mutant compared with the wild type (Extended Data Fig. 2a–e), implying a general reduction of H⁺-ATPase activity in the mutant. TMK1 and TMK4 are functionally redundant in the regulation of the growth of *Arabidopsis* seedlings, as neither of the *tmk1* and *tmk4* single-knockout mutants exhibit a visible growth defect, whereas the *tmk1 tmk4* double mutants show severe growth retardation, especially in hypocotyl elongation[28] (also see below).

We next analysed the phosphorylation status of the penultimate Thr residue by immunoblotting using antibodies against phosphorylated Thr947 (pThr947), which recognize the unique phosphorylation of the penultimate Thr residue in all of the AHA isoforms[27]. Fusicoccin promotes the binding of 14-3-3 to the phosphorylated C-terminal region of PM H⁺-ATPase, resulting in the activation of the pump. As shown previously[14], fusicoccin increased the level of phosphorylation of the penultimate Thr residue in wild type Col-0 seedlings (Extended Data Fig. 2f, g). Similarly, treatments with auxin at micromolar or nanomolar levels increased its phosphorylation levels (Fig. 2a, b and Extended Data Fig. 2f, g). Compared with the untreated wild type, the level of phosphorylation of the penultimate Thr residue was reduced in the *tmk1-1 tmk4-1* mutant (Fig. 2a, b and Extended Data Fig. 2f, g). Importantly, auxin-induced phosphorylation of this residue was nearly abolished in the *tmk1-1 tmk4-1* mutant (Fig. 2a, b and Extended Data Fig. 2f, g). By contrast, fusicoccin treatment still increased the level of phosphorylation of the penultimate Thr residue in the *tmk1-1 tmk4-1* mutant (Extended Data Fig. 2f, g), suggesting that the *tmk1-1 tmk4-1* mutant was able to respond to other stimuli in regulating AHA phosphorylation at the penultimate Thr residue. Thus, TMK1 and TMK4 are required selectively for the auxin-induced increase in phosphorylation of the penultimate Thr residue.

To assess whether TMKs directly phosphorylate AHA at this penultimate Thr residue, we next immunoprecipitated AHA1–GFP from *Arabidopsis* protoplasts that transiently expressed this fusion protein for an in vitro phosphorylation assay. Recombinant TMK1 kinase domain (TMK1[KD]), but not the kinase-dead mutant (TMK1[Km]), greatly increased the phosphorylation of AHA1–GFP at the Thr948 residue in vitro (Fig. 2c, d). We further determined whether TMK1 phosphorylates a synthetic peptide containing the 16 C-terminal amino acid residues from AHA1 (AHA1-C16) in an in vitro assay using the recombinant TMK1[KD]. MS analysis showed that the penultimate Thr residue (Thr15 of the peptide, Thr948 of AHA1) of AHA1-C16 was highly phosphorylated by TMK1[KD], but not by TMK1[Km] (Fig. 2e and Extended Data Fig. 2h). The second to the last Thr residue (T9T15, Thr9 and Thr15 in the AHA1-C16 peptide.) on AHA1-C16 was weakly phosphorylated by TMK1[KD]. Neither TMK1[KD] nor TMK1[Km] phosphorylated a scrambled synthetic peptide (Fig. 2e). Thus, TMK1 specifically phosphorylates the penultimate Thr residue of AHA1. Together with auxin-induced rapid interaction between TMK1 and AHA1 and the requirement of TMK1 and TMK4 for auxin-induced AHA phosphorylation in vivo, these results strongly indicate a role for auxin-activated TMK1 in directly phosphorylating AHA1.

We next investigated whether TMK1 and TMK4 are required for the activation of PM H⁺-ATPase by auxin. The activation of PM H⁺-ATPase couples with the ATP hydrolysis[14]. As shown previously[27], auxin treatment for 30 min increased ATP hydrolysis in the aerial parts of wild-type *Arabidopsis* seedlings by 50% (Fig. 2f). Neither *tmk1-1* nor

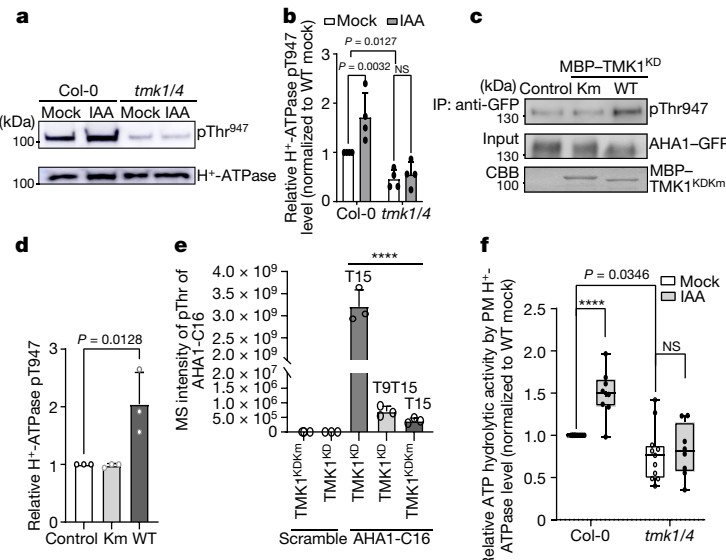

**Fig. 2 | TMK1 and TMK4 are required for auxin-induced phosphorylation and activation of the PM H⁺-ATPase. a**, Western blot detection of phosphorylated H⁺-ATPase in the aerial parts of wild-type and *tmk1-1 tmk4-1* (*tmk1/4*) mutant plants (top, anti-pThr947). AHA protein levels were determined using anti-H⁺-ATPase antibodies (bottom). Seedlings were treated with mock buffer or 10 μM IAA for 30 min. **b**, Quantification of the phosphorylation level of the H⁺-ATPase. Data are mean ± s.d. *n* = 4 independent experiments. **c**, TMK1^KD phosphorylated AHA1–GFP in vitro. TMK1 (TMK1^KD, WT) or the kinase-dead form (TMK1^KDKm) was incubated with protoplast-expressed AHA1–GFP, and its phosphorylation state was determined using anti-pThr947 (pThr947) antibodies. CBB, Coomassie Brilliant Blue. **d**, Quantification of the Thr948 phosphorylation level (determined using anti-pThr947 antibodies) of

the AHA1–GFP. Data are mean ± s.d. *n* = 3 biological replicates. **e**, MS detection of AHA1-C16 phosphorylation by TMK1^KD in vitro. The graph shows the abundance of phosphorylated peptides at the indicated residues analysed by MS. Data are mean ± s.d. *n* = 3. Two biological replicates with three technical replicates each were performed. **f**, Auxin induction of H⁺-ATPase activity was abolished in *tmk1-1 tmk4-1*. The values shown are relative ATP hydrolytic activity of indicated samples to that of mock Col-0. The box bounds the interquartile range divided by the median (central lines), and the Tukey-style whiskers extend to a maximum of 1.5× interquartile range from 25th and 75th percentiles. *n* = 11 (mock) and *n* = 8 (IAA) . Statistical analysis was performed using two-way ANOVA (**b** and **f**) or one-way ANOVA (**d** and **e**); ****P < 0.0001.

*tmk4-1* mutations significantly affected the basal level of ATP hydrolysis or auxin-induced changes in ATP hydrolysis (Extended Data Fig. 2i). However, the basal level of ATP hydrolysis was significantly reduced in the *tmk1-1 tmk4-1* mutant (Fig. 2f), consistent with the reduced level of phosphorylation of the penultimate Thr residue (Fig. 2a, b). Importantly, auxin-induced enhancement of ATP hydrolysis was abolished in this double mutant (Fig. 2f), indicating that TMK1 and TMK4 are essential for auxin-induced H⁺-ATPase activation. In agreement with the compromised H⁺-ATPase activity, compared with the wild type, the *tmk1-1 tmk4-1* mutant was more tolerant to lithium (Extended Data Fig. 2j, k), a toxic alkali cation, the uptake of which is coupled with the activation of H⁺-ATPase and PM hyperpolarization. In particular, the aerial part of wild-type seedlings became chlorotic, whereas the aerial part of the *tmk1-1 tmk4-1* seedlings remained green after growth on lithium. This is in contrast to SAUR19-OX lines, which display increased H⁺-ATPase activity and therefore exhibit much higher sensitivity to lithium[17]. These findings together demonstrate that TMK1 and TMK4 are required for auxin-induced PM H⁺-ATPase activation.

To assess the consequences of the reduced PM H⁺-ATPase activation in *tmk1-1 tmk4-1* plants, we introduced membrane-impermeable 8-hydroxypyrene-1,3,6-trisulfonic acid trisodium salt (HPTS) as a ratiometric fluorescent pH indicator for assessing changes in the apoplastic pH at a cellular resolution in *Arabidopsis thaliana* hypocotyls[4]. Two different forms of HPTS (the protonated and deprotonated forms) were visualized in two independent channels with excitation wavelengths of 405 nm and 458 nm, respectively. The apoplastic pH correlates with the ratiometric values (signal intensity from the 458 nm channel divided by that from the 405 nm channel)[4,33]. As a positive control for the HPTS-based pH indicator, we monitored the apoplastic pH in hypocotyls of the *ost2-2D* mutant harbouring the constitutively activated AHA1 (ref. [34]). As shown previously[4], the *ost2-2D* mutant exhibited

lower 458/405 values compared with the wild type (Fig. 3a, b), confirming its enhanced apoplastic acidification. By contrast, significantly higher 458/405 values were observed in *tmk1-1 tmk4-1* hypocotyls, suggesting apoplastic alkalization in the mutant (Fig. 3a, b). Furthermore, the apoplastic pH of the *tmk1-1 tmk4-1* mutant was restored to the wild-type level when this mutant was complemented with wild-type TMK1 (Extended Data Fig. 3a, b), indicating that TMK1 is essential for the regulation of the apoplastic pH.

Importantly, hypocotyl cell length was correlated with the pH value of the mutant when compared with the wild type (Fig. 3c). In *tmk1-1 tmk4-1* mutants, the mean length of hypocotyl cells was significantly shorter than in the wild type[28] (Figs. 3a, c), and the hypocotyl cell lengths were

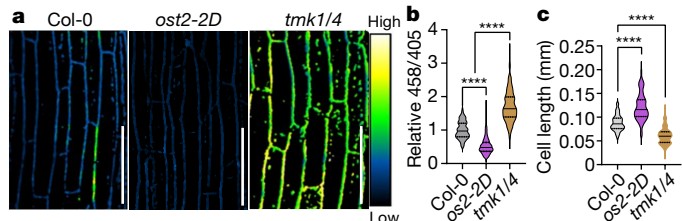

**Fig. 3 | TMK1 and TMK4 are required for apoplastic acidification and cell elongation in *Arabidopsis* hypocotyl. a**, **b**, Comparison (imaging (**a**) and quantification (**b**)) of the apoplastic pH in wild type (Col-0), *ost2-2D* and *tmk1-1 tmk4-1* plants. Changes in pH were visualized with ratiometric values of fluorescent HPTS. The mean 458/405 values of *ost2-2D* and the *tmk1-1 tmk4-1* mutant relative to the WT are shown on the *y* axis. *n* = 300 (6 hypocotyls, 50 cells for each). **c**, Epidermal cell lengths of hypocotyls. *n* = 140 (Col-0), *n* = 64 (*ost2-2D*) and *n* = 134 (*tmk1-1 tmk4-1*) cells. Statistical analysis was performed using one-way ANOVA (**b** and **c**) (three independent assays); ****P ≤ 0.0001. For **a**, scale bars, 100 μm.

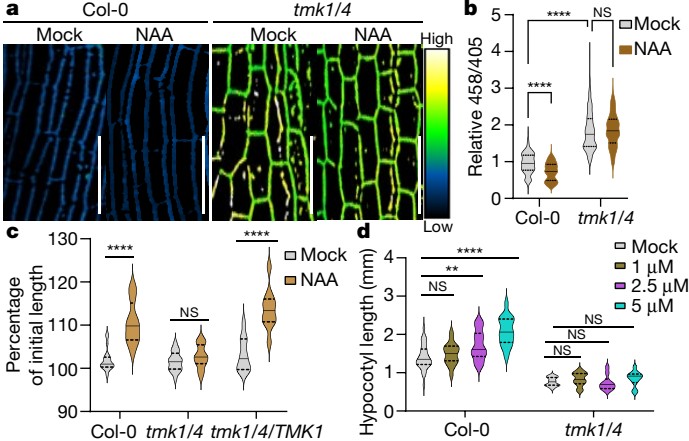

**Fig. 4 | TMK1 and TMK4 are required for auxin-induced apoplastic acidification and hypocotyl elongation. a**, The effect of auxin (100 nM NAA for 15 min) on the apoplastic pH changes visualized by HPTS staining. **b**, Quantitative analysis of **a**. $n = 350$ (7 hypocotyls, 50 cells for each). **c**, Auxin-induced rapid elongation hypocotyl segments from wild-type, *tmk1-1 tmk4-1* (*tmk1/4*) and *tmk1-1 tmk4-1 TMK1* (*tmk1/4/TMK1*) plants. The relative length when comparing the hypocotyl segments at 30 min to that at 0 min is represented on the *y* axis. The results were analysed using two-way ANOVA. $n = 20$ hypocotyl sections per line. **d**, Auxin-induced hypocotyl elongation in seedlings. $n = 16$ (Col-0) and $n = 10$ (*tmk1-1 tmk4-1*) (three independent assays). Statistical analysis was performed by two-way ANOVA (**b**–**d**). **$P \le 0.01$, ****$P \le 0.0001$; NS, not significant. For **a**, scale bars, 100 μm.

largely restored when the mutant was complemented with wild-type TMK1 (Extended Data Fig. 3c). By contrast, increased apoplastic acidification is linked to an increase in cell length and hypocotyl length in *ost2-2D* plants (Fig. 3a, c and Extended Data Fig. 3d, e).

Auxin promotes apoplastic acidification in hypocotyls through the activation of PM H[+]-ATPase, contributing to auxin-induced cell elongation[16,27]. We found that the auxin-induced acidification in the apoplast was abolished in *tmk1-1 tmk4-1* plants (Fig. 4a, b), suggesting an essential role of TMK1 and TMK4 in auxin-triggered PM H[+]-ATPase activation. Moreover, exogenous NAA promoted the rapid elongation of auxin-depleted hypocotyl segments (Fig. 4c) and the elongation of whole hypocotyls (Fig. 4d) from wild-type seedlings, but not from the *tmk1-1 tmk4-1* seedlings. The *tmk1-1 tmk4-1* mutant complemented with TMK1 exhibited a normal response to auxin in promoting hypocotyl segment elongation (Fig. 4c). The severe defect in *tmk1-1 tmk4-1* hypocotyl elongation was partially rescued when *tmk1-1 tmk4-1* seedlings were grown on medium with a lower pH (pH 5.0 and pH 4.3) compared with standard medium (pH 5.7) (Extended Data Fig. 4a, b). Moreover, *ost2-2D*, which caused activation of the PM H[+]-ATPase, partially rescued the hypocotyl elongation defect of *tmk1-1 tmk4-1* mutants (Extended Data Fig. 4c, d). TMK1 and TMK4 probably activate other downstream pathways to regulate hypocotyl elongation in addition to the PM H[+]-ATPase activation, such as ROP GTPase signalling to the organization of the cytoskeleton[20,35]. Such additional downstream pathways may explain the incomplete rescue of the hypocotyl elongation defect in *tmk1-1 tmk4-1* plants by *ost2-2D*. Together, our results indicate that TMK1 and TMK4 are required for apoplastic acidification through auxin-triggered activation of PM H[+]-ATPase, contributing to auxin regulation of hypocotyl cell elongation.

In this Article, we show that TMK1 directly interacts with PM H[+]-ATPases on the PM, and this interaction was induced rapidly (within 10 s) by auxin treatment (Fig. 1e, f), well preceding an auxin-induced increase in cell elongation[16]. Thus, the auxin-induced TMK–AHA association can be considered to be the very early response for auxin signal transduction. Our results suggest that, once interacting with AHAs after auxin stimulation, TMK1 directly phosphorylates AHA1 at the penultimate Thr residue (Fig. 2 and Extended Data Fig. 2). An accompanying paper by Li et al. shows that this auxin-induced phosphorylation of AHA's penultimate Thr residue occurred in root tissues within 2 min after auxin treatment[36], nearly as rapid as the auxin-induced interaction between TMK1 and AHA1 (Fig. 1b, f). TMKs therefore regulate AHA

activation by affecting the phosphorylation status of the penultimate Thr residue. As a consequence, auxin induced apoplastic acidification in a TMK1/TMK4-dependent manner in hypocotyl cells (Figs. 3a, b and 4a, b). Moreover, reducing the apoplast pH either genetically by *ost2-2D* or growing seedlings in an acidic environment partially restored the hypocotyl elongation defect of *tmk1-1 tmk4-1* plants (Extended Data Fig. 4a–d). These data suggest that, after activation by auxin, the cell-surface auxin-signalling components TMKs act as protein kinases to directly and rapidly initiate the phosphorylation and activation of PM H[+]-ATPase, although our findings do not exclude the possibility that the TMK-mediated AHA phosphorylation may also respond to other stimuli under certain conditions. By contrast, the nuclear auxin signalling inhibits ATPase dephosphorylation through the TIR1/AFB–SAUR-PP2C.D pathway[17,37,38]. Thus, the current findings support the hypothesis that the cell-surface and intracellular auxin-signalling pathways, respectively, initiate and sustain PM H[+]-ATPase activation in cells in which auxin promotes cell expansion, such as in hypocotyls, and collectively explain the acid growth theory. In roots, TMK-dependent auxin signalling also promotes ATPase activation, but to counter the rapid alkalization (or membrane depolarization) activated by TIR1/AFBs[36,39,40]. Importantly, these findings, together with the recent findings on the TMK-mediated noncanonical auxin signalling in regulating pavement cell morphogenesis[18,41], differential growth of the apical hook[18], lateral root formation[19], root gravitropic response[42] and thermomorphogenesis[23], are emerging as a common theme that auxin regulates growth and developmental processes through the coordinate actions of intracellular and cell-surface auxin-signalling systems.

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

# Methods

## Plant materials and growth conditions

*A. thaliana* Columbia ecotype (Col-0) was used as the wild type in this study. *ost2-2D* seeds were obtained from J. Leung. The *tmk1-1 tmk4-1* mutant in the Col-0 background and *pTMK1-TMK1-GFP* transgenic lines (in the *tmk1-1 tmk4-1* mutant) were described previously[18,19]. The *ost2-2D tmk1-1 tmk4-1* mutants were generated by genetic crosses and confirmed by genotyping. *Arabidopsis* plants were grown in soil (Sungro S16-281) in a growth room at 23 °C, 40% relative humidity and 75 µE m$^{-2}$ s$^{-1}$ light under a 12 h photoperiod for approximate 4 weeks before protoplast isolations. To grow *Arabidopsis* seedlings, the seeds were surface-sterilized with 50% bleach for 10 min (*tmk1-1 tmk4-1* seeds were sterilized with 75% (v/v) ethanol for 5 min), and washed three times with sterilized distilled H$_2$O, and then placed onto plates with 1/2 MS medium containing 0.5% sucrose and 0.8% agar at pH 5.7 in the dark with vertical growth. Then 2–3 d after germination, hypocotyls were used for cell characterization.

## Plasmid construction and generation of transgenic plants

Full-length and truncated variants *TMK1*, *AHA1* and *AHA2* were amplified by PCR from Col-0 cDNA and cloned into a protoplast transient expression vector (HBT vectors obtained from L. Shan and P. He) or the plant binary vectors pGWB641 and pGWB644. Stable transgenic lines were generated using standard *Agrobacterium tumefaciens*-mediated transformation in the *tmk1-1 tmk4-1* mutant or Col-0 (ref. [43]). The full-length cDNAs of *TMK1* and *AHA1* were amplified by PCR, and then cloned into the pDONR221-P1P4 and pDONR221-P3P2 vectors using the BP recombination reaction (Invitrogen), respectively. pDONR221-P1P4-TMK1 was recombined with pDONR221-P3P2-AHA1 into pFRETgc-2in1-NC to generate pFRET-mEGFP-AHA1+TMK1-mCherry[44]. pDONR221-P3P2-AHA1 was recombined with pENTRL1-pLac-LacZalpha-L4 (Invitrogen) into pFRETgc-2in1-NC to generate pFRET-mEGFP-AHA1. pDONR221-P1P4-TMK1 was recombined with pENTRL3-pLac-Tet-L2 (Invitrogen) into pFRETgc-2in1-CC to generate pFRET-MK1-mCherry. The *AHA2* C-terminal region was cloned into pDest-565, and expressed in *Escherichia coli* (Rosetta, BL21) (a list of the primers is provided in Supplementary Table 1).

## Determination of H$^+$-ATPase phosphorylation levels

The immunoblot analysis was performed as described by Hayashi[45] using specific antibodies against the catalytic domain of AHA2 and phosphorylated Thr947 in AHA2 (1:5,000 dilution)[45]. These antibodies recognize not only AHA2 but also other H$^+$-ATPase isoforms in *Arabidopsis*[46]. In brief, the roots were removed from 1/2 MS-grown seedlings (aged 5 d), and the remaining aerial sections were incubated in a KPSC buffer (10 mM potassium phosphate, pH 6.0, 2% sucrose, 50 µm chloramphenicol) in the dark for 10 h. The buffer was then replaced every hour. The pretreated tissues were incubated in the presence of 100 nM IAA for 10 min or 10 µM IAA for 30 min in the dark. The aerial sections were collected and grounded with a plastic pestle, followed by solubilization in 40 µl of SDS buffer (3% (w/v) SDS, 30 mM Tris-HCl (pH 8.0), 10 mM EDTA, 10 mM NaF, 30% (w/v) sucrose, 0.012% (w/v) Coomassie Brilliant Blue and 15% (v/v) 2-mercaptoethanol), and the homogenates were centrifuged at room temperature (10,000g for 5 min). Next, 12 µl of the supernatant was loaded onto 10% (w/v) SDS–PAGE gels to assess the H$^+$-ATPase or the phosphorylated penultimate Thr levels using the respective above-mentioned antibodies. Goat anti-rabbit IgG (1:10,000 dilution) conjugated to horseradish peroxidase (Santa Cruz, sc-2357) was used as a secondary antibody. The chemiluminescent signal was quantified using ImageJ (Fiji, Java 1.8.0_172) (Fig. 2a, Extended Data Fig. 2f and Supplementary Fig. 1).

## HPTS staining and imaging

HPTS staining and imaging were performed as described by Barbez[4] with modifications. In brief, two-day etiolated seedlings were transferred and incubated with 1 mM HPTS (from 100 mM water stock) with 0.01% Triton X-100 under vacuum (10–15 pa) 5 min. The seedlings were then incubated with HPTS for 60 min in the liquid growth medium. The seedlings were subsequently mounted in the same growth medium on a microscopy slide and covered with a coverslip. For auxin treatment, seedlings were incubated in 1/2 MS growth medium supplemented with 1 mM HPTS and NAA in the stated concentration for 15 min and subsequently mounted in the same growth medium on a microscopy slide and covered with a coverslip. Seedling imaging was performed using an inverted Zeiss 880 confocal microscope equipped with a highly sensitive GaAsP detector. Fluorescent signals for the protonated HPTS form (excitation, 405 nm; emission peak, 514 nm), as well as the deprotonated HPTS form (excitation, 458 nm; emission peak, 514 nm), were detected using a ×10 water-immersion objective.

## IP–MS analyses

The *pTMK1::gTMK1-GFP/tmk1-1 tmk4-1* seedlings were grown on 1/2 MS medium for 10 d, and the entire seedlings were collected and ground in liquid nitrogen with a mortar and pestle. Total proteins were extracted using extraction buffer (50 mM Tri-HCl pH 7.4, 150 mM NaCl, 5 mM EDTA, 0.5% Triton X-100 with protease inhibitor and phosphatase inhibitor) on ice. The extracts were centrifugated at 13,000g for 30 min, and the supernatants were incubated with GFP-Trap agarose beads (GFP-Trap_A, gta-20, ChromoTek) at 4 °C for 2 h to immunoprecipitate TMK1–GFP proteins. The agarose beads were washed and resuspended with 50 mM Tris-Cl buffer (pH 7.8). One tenth of the beads was used for immunoblot analysis with anti-GFP antibodies. The remaining agarose beads were used for LC coupled with tandem MS (LC–MS/MS) analysis. MS analysis was carried out by Orbitrap Fusion mass spectrometry (Thermo Fisher Scientific).

## Phosphoproteomics analyses

Col-0 and the *tmk1-1 tmk4-1* seedlings were cultured on 1/2 MS plate for 5 d, then the aerial parts of seedlings were transferred to 1/2 MS liquid medium and incubated in the KPSC buffer (10 mM potassium phosphate, pH 6.0, 2% sucrose and 50 µm chloramphenicol) in the dark overnight, and the buffer was replaced every 1 h for 12 h (ref. [47]). Seedlings were collected and flash-frozen in liquid nitrogen. A total of 1 g of frozen shoots (fresh weight) was ground with a liquid-nitrogen-precooled mortar and pestle, and then homogenized in 5 ml extraction buffer (50 mM Tris-HCl buffer (pH 8), 0.1 M KCl, 30% sucrose, 5 mM EDTA and 1 mM dithiothreitol (DTT) in Milli-Q water, 1× complete protease inhibitor mixture and the PhosSTOP phosphatase inhibitor mixture) in a Dounce Homogenizer. At least 50 strokes were performed. The homogenate was filtered through four layers of miracloth and centrifuged at 5,000g at 4 °C for 10 min. Half of the supernatant was used to resuspend the pellet, and the mixture was centrifuged again at 5,000g 4 °C for 10 min. The two fractions of the supernatants were combined and mixed with 3, 1 and 4 volumes of methanol, chloroform and water, respectively. The mixtures were centrifuged at 5,000g for 10 min, and the aqueous phase was removed. After the addition of four volumes of methanol, the proteins were pelleted by centrifugation at 4,000g for 10 min. The pellets were washed with 80% acetone and resuspended in 6 M guanidinium hydrochloride in 50 mM triethylammonium bicarbonate buffer (pH 8). The proteins were used for tandem mass tag labelling according to the Kit protocol (Thermo Fisher Scientific, 90096) and quantification by MS.

Nano-LC–MS/MS was performed using the Dionex rapid-separation liquid chromatography system interfaced with a QExactive HF (Thermo Fisher Scientific). Samples were loaded onto an Acclaim PepMap 100 trap column (75 µm × 2 cm, Thermo Fisher Scientific) and washed with buffer A (0.1% trifluoroacetic acid) for 5 min with a flow rate of 5 µl min$^{-1}$. The trap was brought in line with the nano analytical column (nanoEase, MZ peptide BEH C18, 130 A, 1.7 µm, 75 µm×20 cm, Waters) with a flow rate of 300 nl min$^{-1}$ with a multistep gradient (4–15% buffer

B (0.16% formic acid and 80% acetonitrile) in 20 min, then 15–25% B in 40 min, followed by 25–50% B in 30 min). MS data were acquired using a data-dependent acquisition procedure with a cyclic series of a full scan acquired with a resolution of 120,000 followed by MS/MS scans (33% collision energy in the HCD cell) with a resolution of 45,000 of the 20 most intense ions with a dynamic exclusion duration of 20 s.

All LC–MS data were analysed using Maxquant (v.1.6.2.6) with the Andromeda search engine. The type of LC–MS run was set to reporter ion MS2 with 10 plex tandem mass tags as isobaric labels. Reporter ion mass tolerance was set at 0.003 Da. LC–MS data were searched against TIAR10 with the addition of potential contaminants. Protease was set as trypsin/P, allowing two misscuts for total proteomic data and three misscuts for the phosphor-enriched sample (post-translational modification sample). Carbamidomethylation of cysteine was set as a fixed modification, N-terminal acetylation, oxidation at methionine as well as phosphorylation at serine, threonine and tyrosine were set as variable modifications. Proteins with a false-discovery rate of <1% were reported. For quantification, spectra were filtered by minimum reporter PIF set at 0.6.

The results were further analysed using Perseus (v.1.6.1.3). The protein group results were first filtered for reverse and contaminant hits and the reporter ion intensity values were further $\log_2$-transformed and normalized to the column total. For group comparisons, statistical significance between groups was analysed using Student's $t$-tests with equal variance on both sides, requiring two valid values in total, and the $Q$ value was calculated using the permutation test. S0 (ref. [48])[2] was set to 1.

The phospho(STY)-site reporter ions were first normalized to proteome column total, and then filtered for reverse and contaminant hits, and the reporter ion intensity values were further $\log_2$-transformed. Each phospho(STY) site was further normalized to protein abundance from proteome data if available. Only sites that have site localization confidence > 75% were included in the analysis. For group comparison, statistical significance between groups was analysed using Student's $t$-tests with equal variance on both sides, requiring two valid values in total, and the $Q$ value was calculated using the permutation test. S0 was set to 1.

### Co-IP assays with transgenic plants

Approximately 1 g of *35S::GFP* and *35S::GFP-AHA1* plants (aged 4 weeks) was ground in liquid $N_2$ and further ground in 0.5 ml of ice-cold co-IP buffer (10 mM HEPES at pH 7.5, 100 mM NaCl, 1 mM EDTA, 10% glycerol, 0.1% Triton X-100 and protease inhibitor mixture from Roche). The homogenates were centrifuged at 12,470$g$ at 4 °C for 10 min. The supernatant was incubated with anti-GFP-Trap antibodies (Chromotek, 3h9, 1:1,000 dilution) for 2 h with gentle shaking. The beads were collected and washed three times with washing buffer (10 mM HEPES (pH 7.5), 100 mM NaCl, 1 mM EDTA, 10% glycerol and 0.1% Triton X-100) and once with 50 mM Tris-Cl (pH 7.5), and analysed by western blotting using anti-TMK1 or anti-TMK4 antibodies[18]. The total input proteins were determined using anti-GFP antibodies (Fig.1a, Extended Data Fig. 1b and Supplementary Fig. 1).

### Co-IP analysis of the TMK–AHA interaction induced by auxin treatment in *Arabidopsis* protoplasts

Wild-type Col-0 leaf protoplasts were prepared as described by Yoo et al.[49]. Protoplasts ($2 \times 10^5$) were co-transfected with *35S::AHA1-HA* or *35S::TMK1-Myc* plasmid DNA, which was prepared using the Invitrogen PureLink Plasmid Maxiprep Kit, and incubated at room temperature for 10 h. Transfected protoplasts were then collected in 2 ml Eppendorf tubes and centrifuged at 100$g$ for 1 min. The supernatant was discarded, and the protoplasts were resuspended with 100 µl W5 solution (2 mM MES-KOH, pH 5.7, 5 mM KCl, 154 mM NaCl and 125 mM CaCl$_2$). The protoplasts were treated with 1 µM NAA for the indicated time periods, frozen in liquid $N_2$ immediately and stored in −80 °C. The samples were lysed

with 0.5 ml of extraction buffer (10 mM HEPES (pH 7.5), 100 mM NaCl, 1 mM EDTA, 10% (v/v) glycerol, 0.5% Triton X-100 and protease inhibitor mixture from Roche). After vortexing vigorously for 30 s, the samples were centrifuged at 12,470$g$ for 10 min at 4 °C. The supernatant was incubated with anti-Myc antibodies (Sinobiological, 100029-MM08, 1:1,000 dilution) for 2 h, and then incubated with protein G agarose beads (Pierce, 20397) for another 2 h at 4 °C with gentle shaking. Beads were collected by centrifugation at 100$g$ for 1 min at 4 °C followed by washing twice with extraction buffer. The beads were washed with 50 mM Tris-Cl buffer (pH 7.5), and the immunoprecipitated proteins were analysed by immunoblotting with anti-HA–HRP antibodies (Invitrogen, 26183, 1:2,000 dilution) (Fig.1b and Supplementary Fig. 1).

### In vitro pull-down assay

MBP- or GST-fusion proteins were expressed in *E. coli* and affinity-purified using standard protocols. In brief, 200 ml of isopropyl-β-D-thiogalactoside-induced cell culture pellet was lysed in 20 ml lysis buffer (containing 0.5% Triton X-100) by sonication on ice. Centrifuge lysates were cleared by centrifuging at 10,000$g$ for 30 min at 4 °C. The supernatant was then incubated with 100 µl amylose resins or glutathione-Sepharose beads at 4 °C for 4 h with gentle rotation. The beads were then centrifuged and washed three times with lysis buffer. Proteins were eluted with GST (10 mM reduced glutathione in 50 mM Tris pH 8.0) or MBP (20 mM Tris-HCl, 200 mM NaCl, 1 mM EDTA, 1 mM DTT, 10 mM maltose, pH 7.4) buffer. The protein concentration was estimated using the NanoDrop ND-1000 spectrophotometer and confirmed using the Bio-Rad Quick Start Bradford Dye Reagent. GST- or GST-fusion proteins (10 µg; immobilized on glutathione-Sepharose beads) were incubated with 10 µg prewashed MBP or MBP fusion proteins at 4 °C in 150 µl of incubation buffer (10 mM HEPES (pH 7.5), 100 mM NaCl, 1 mM EDTA, 10% glycerol and 0.5% Triton X-100) for 1 h. The beads were collected and washed three times with washing buffer (20 mM HEPES (pH 7.5), 300 mM NaCl, 1 mM EDTA and 0.5% NP-40) and once with 50 mM Tris-HCl (pH 7.5). Proteins in the beads were analysed by immunoblotting with anti-GST (Santa Cruz, sc-138, 1:1,000 dilution) or anti-MBP (Invitrogen, PA1-989, 1:1,000 dilution) antibodies (Extended Data Fig. 1d and Supplementary Fig. 1).

### Vanadate-sensitive ATPase activity measurement

ATP hydrolysis by PM H$^+$-ATPase was measured in a vanadate-sensitive manner as previously described[27]. In brief, the aerial parts of seedlings (Col-0, *tmk1-1*, *tmk4-1*, and *tmk1-1 tmk4-1*; aged 14 d) were incubated in KPSC buffer (10 mM potassium phosphate, pH 6.0, 2% sucrose, 50 µm chloramphenicol) in the dark for 10 h. The buffer was replaced every hour. The pretreated tissues were incubated in the presence of 10 µM IAA for 30 min in darkness. The tissues were homogenized with homogenization buffer (50 mM MOPS-KOH, pH 7.0, 100 mM KNO$_3$, 2 mM sodium molybdate, 0.1 mM NaF, 2 mM EGTA, 1 mM phenylmethyl-sulfonyl fluoride and 20 µM leupeptin) and the homogenates were centrifuged at 10,000$g$ for 10 min; the obtained supernatant was further ultracentrifuged at 45,000$g$ for 60 min. The resultant pellet (the microsomal fraction) was resuspended in the homogenization buffer. The ATP hydrolytic activity of the microsomal fraction was measured in a vanadate-sensitive manner, and the inorganic phosphate released from ATP was measured[27].

### In vitro phosphorylation

Protoplasts were isolated from plants expressing AHA1–GFP as described above. Agarose-immobilized (GFP-Trap beads, Chromotek, gta-100) AHA1–GFP proteins were incubated with 1 µg MBP–TMK1$^{KD}$ or MBP–TMK1$^{KDKm}$ recombinant proteins (expressed in *E. coli* and isolated by affinity purification) in phosphorylation buffer (5 mM HEPES, 10 mM MgCl$_2$, 10 mM MnCl$_2$, 1 mM DTT and 50 µM ATP)[18] at room temperature (24 °C) for 1 h. After incubation, the reaction was stopped by adding 4× SDS loading buffer. Proteins in the beads were analysed by

immunoblotting using anti-pT947, anti-AHA1-cat or anti-GFP (Chromotek, 3h9) antibodies (Fig. 2c and Supplementary Fig. 1).

For in vitro phosphorylation of synthetic peptides, 10 mg synthetic peptide AHA1-C16 (KLKGLDIDTAGHHYTV) or scrambled peptide (GDAHVKITHLDKGLIT) was incubated with 1 μg MBP–TMK1[KD] or MBP–TMK1[K-DKm] recombinant proteins in phosphorylation buffer for 1 h. The peptide mixtures were then analysed using MS.

### Auxin-induced rapid hypocotyl segment elongation

For analysing auxin-induced elongation of hypocotyl segments, the auxin-depleted hypocotyl sections (2 mm) were transferred to growth medium (10 mM KCl, 1 mM MES-KOH, pH 5.7, 0.8% agar) with/without 10 μM NAA for 30 min. The hypocotyl sections were photographed and measured using ImageJ at 0 min and 30 min after treatments. For analysing auxin-induced hypocotyl elongation in seedlings, 1/2 MS-grown seedlings (aged 4 d) were transferred into 1/2 MS medium containing the indicated concentrations of NAA and incubated for additional 48 h under normal growth conditions. Hypocotyl lengths were measured using ImageJ after treatment.

### Fabrication of the protoplast-capture microfluidics chip

The design parameters for the protoplast capture chip (Extended Data Fig. 1e) are as follows: each capture unit consists of three pillars. The average diameter of each pillar is 50 μm. The distance between the two entrance pillars is 52 μm. The shortest distance between the entrance pillar and the bottom pillar is 27 μm. The height of the capture chamber is 80 μm. Each chip contains an array of 1,866 capture units. A master mould was fabricated on a silicon wafer using the traditional photolithography technique by Beijing Borui Yisheng Technology. In brief, SU-8 3050 was centrifuged at 1,600 r.p.m., soft-baked at 95 °C for 30 min and exposed at 260 mJ cm$^{-2}$. After exposure, the wafer was post-exposure baked at 95 °C for 5 min, developed for 3 min and air-dried with pressurized nitrogen.

The protoplast capture chip was fabricated with soft lithography using polydimethylsiloxane (PDMS) (Sylgard 184 Silicone Elastomer Kit, The DOW Chemical Company). In brief, PDMS prepolymer and curing agent were mixed at a 10:1 ratio, degassed in a vacuum chamber with negative pressure, poured onto the master mould and baked at 80 °C for 2 h. After curing, the PDMS slab was peeled off, hole-punched and, finally, plasma-oxidized to adhere to the cover glass.

### Auxin treatment and FRET analysis of protoplasts captured in microfluidics chips

*Arabidopsis* (Col-0) protoplasts were isolated and transfected with 2in1 AHA1–GFP, TMK1–mCherry and AHA1–GFP/TMK1–mCherry vectors as described. Before the experiment, the protoplast capture chip was filled with a protoplast-suspending WI solution (4 mM pH 5.7 MES-KOH, 0.5 M mannitol and 20 mM KCl). After air bubbles were entirely removed from the chip, the protoplast suspension was injected slowly into the chip from inlet 1 (Fig. 1e). The time-lapse FRET images were acquired at 5 s per frame using the Zeiss LSM880 confocal laser scanning microscope (argon 488 30%, and argon 561 3%). NAA or mock solution was injected into the chip from inlet 2 (Fig. 1e) at 25 s after the live imaging started.

The FRET efficiency was analysed using FRET sensitized emission methods[46]. In brief, the AHA1–GFP only, TMK1–mCherry only sample and the FRET samples were imaged using the same microscope settings (the donor and FRET channels were excited with 30% argon 488 nm, and the emissions were collected using 498–551 nm and 600–670 nm, respectively; the acceptor channel was excited with 3% argon 561 nm and the emissions were collected using 600–670 nm). To avoid interference by chlorophyll autofluorescence, protoplasts with concentrated chloroplasts at one side of the cell were processed for quantification. A segmented line was drawn along the PM region opposite to the site of chloroplasts to measure the mean signal intensity for each channel using Image-Pro Plus (http://www.mediacy.com/imageproplus) and LAS-X (Leica). The correction factors $\beta$, $\alpha$, $\gamma$ and $\delta$ were calculated with the donor- and acceptor-only reference samples, then the FRET efficiency was calculated using the equation below[50].

$$E_{(\text{FRET}-\text{SE})} = \frac{\text{FRET} - \text{donor} \times \beta - \text{acceptor} \times (\gamma - \alpha \times \beta)}{\text{acceptor} \times (1 - \beta \times \delta)}$$

The mean FRET efficiency and s.d. from 10 cells of 100 nM NAA or mock treatment are presented in Fig. 1g. To generate the FRET efficiency heat-map image, the plasma membrane region in the side opposite to the chloroplasts was cropped as the region of interest to avoid autofluorescence (Extended Data Fig. 1d). The cropped images from the donor, FRET and acceptor channels were processed using the image calculator module of ImageJ with the $E_{(\text{FRET-SE})}$ equation shown above.

### Reporting summary

Further information on research design is available in the Nature Research Reporting Summary linked to this paper.

## Data availability

Data supporting the findings of this study are available within the paper and its Supplementary Information. Mass spectrometry raw data are available at the MassIVE under accession number msv000087822. Source data are provided with this paper.

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

**Acknowledgemacents** We thank the members of the Yang laboratory for discussion and comments on this work; J. Leung (Department of Institute Jean-Pierre Bourgin, INRA) for *ost2-2D* seeds; K. Iba (Kyushu University) for *35S::GFP-AHA1* seeds; C. Grefen (University of Tubingen) for FRET analyses 2in1 binary vectors; and L. Shan (University of Texas A&M) for protoplast transient expression vectors. This work was in part supported by an NIH grant (GM100130) to Z.Y.; K.T. (20K06685) and T.K. (20H05687 and 20H05910) were funded by MEXT/JSPS KAKENHI. W.M.G. was funded by NIH (GM067203). H.Z. was funded by NIH (S10OD016400).

**Author contributions** W.L. and Z.Y. conceived the project and designed the experiments. W.L. conducted most of the experiments, with contributions from X.Z., W.T., K.T., X.P., J.D., H.R. and W.M.G.; W.L., S.P. and H.Z. conducted and analysed all the MS analyses of this paper; X.Z., J.D., X.P. and X.Y.Z. designed and conducted the FRET analyses. K.T. and T.K. conducted and analysed the ATPase activities. W.L., Z.Y. and W.M.G. wrote the manuscript with input from all of the other authors.

**Competing interests** The authors declare no competing interests.

**Additional information**
**Correspondence and requests for materials** should be addressed to Zhenbiao Yang.

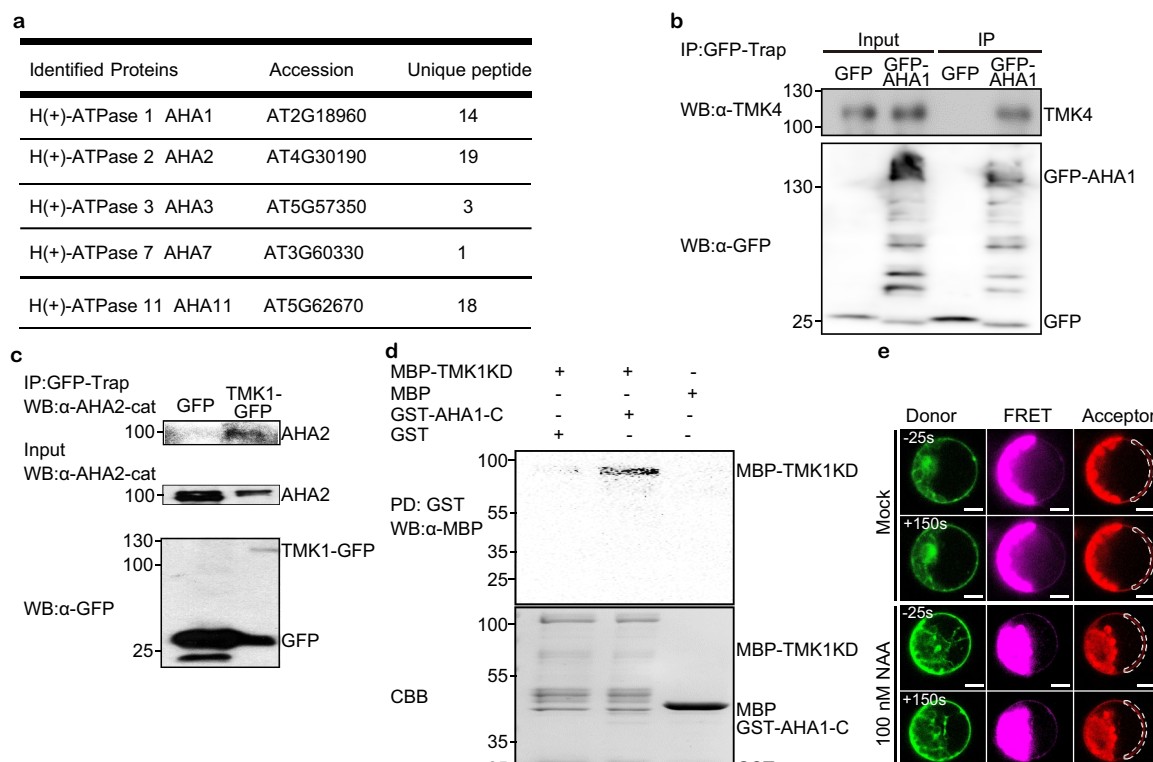

**Extended Data Fig. 1 | TMK interacts with AHAs *in planta* and *in vitro*.**
**a**, Summary of LC-MS/MS analysis of AHAs associated with TMK1-GFP. The number of unique AHA peptides identified in the immunoprecipitates from *pTMK1::TMK1-GFP* transgenic seedlings is shown. IP-MS did not identify any AHA peptides from control *pTMK1::GFP* seedlings. **b**, TMK4 associates with AHA1 in transgenic plants. The membrane proteins from 4-week-old *35S::GFP* only and *35S::GFP-AHA1* plants were immunoprecipitated with α-GFP-Trap antibody and analysed with Western blots using an α-TMK4 antibody (Top). The expression of GFP-AHA1 and GFP control in transgenic plants is shown (Bottom). **c**, TMK1 associates with AHA2 in transgenic plants. Membrane proteins from 4-wk-old *35S::GFP* and *pTMK1::TMK1-GFP/tmk1-1/4-1* transgenic plants were immunoprecipitated with α-GFP-Trap antibody and analysed with Western blots using an α-AHA2 antibody (Top). The expression of TMK1-GFP and GFP control in transgenic plants is shown (Bottom). **d**, TMK1's cytoplasmic kinase domain (KD) interacts with AHA2's C-terminal domain *in vitro*. *E. coli*-expressed maltose-binding protein (MBP)-TMK1KD or MBP proteins were incubated with glutathione bead-bound glutathione-S-transferase (GST)-AHA2-C or GST (Pull-down:GST), and the beads were collected and washed for Western blotting of immunoprecipitated proteins with α-MBP antibody (left). The input GST-AHA2-C, MBP-TMK1KD, MBP, and GST proteins were detected by Coomassie brilliant blue staining (CBB). **e**, Representative confocal images of Arabidopsis protoplasts expressing TMK1-mCherry (FRET acceptor) and AHA1-GFP (FRET donor) used for FRET analysis. Shown are images collected before (−25 sec) and after (+150 sec) auxin treatment for three channels: Donor (excitation: 488 nm, emission: 498-551 nm), FRET (excitation: 488 nm, emission: 600-670) and acceptor (excitation: 561 nm, emission: 600-670 nm). These images are used for FRET efficiency analysis shown in Fig. 1e. To avoid autofluorescence from chorophylls, only the PM region (dotted lines) away from chloroplasts was selected for FRET analysis. Scale bar, 10 μm. 3 independent analyses were conducted with similar results.

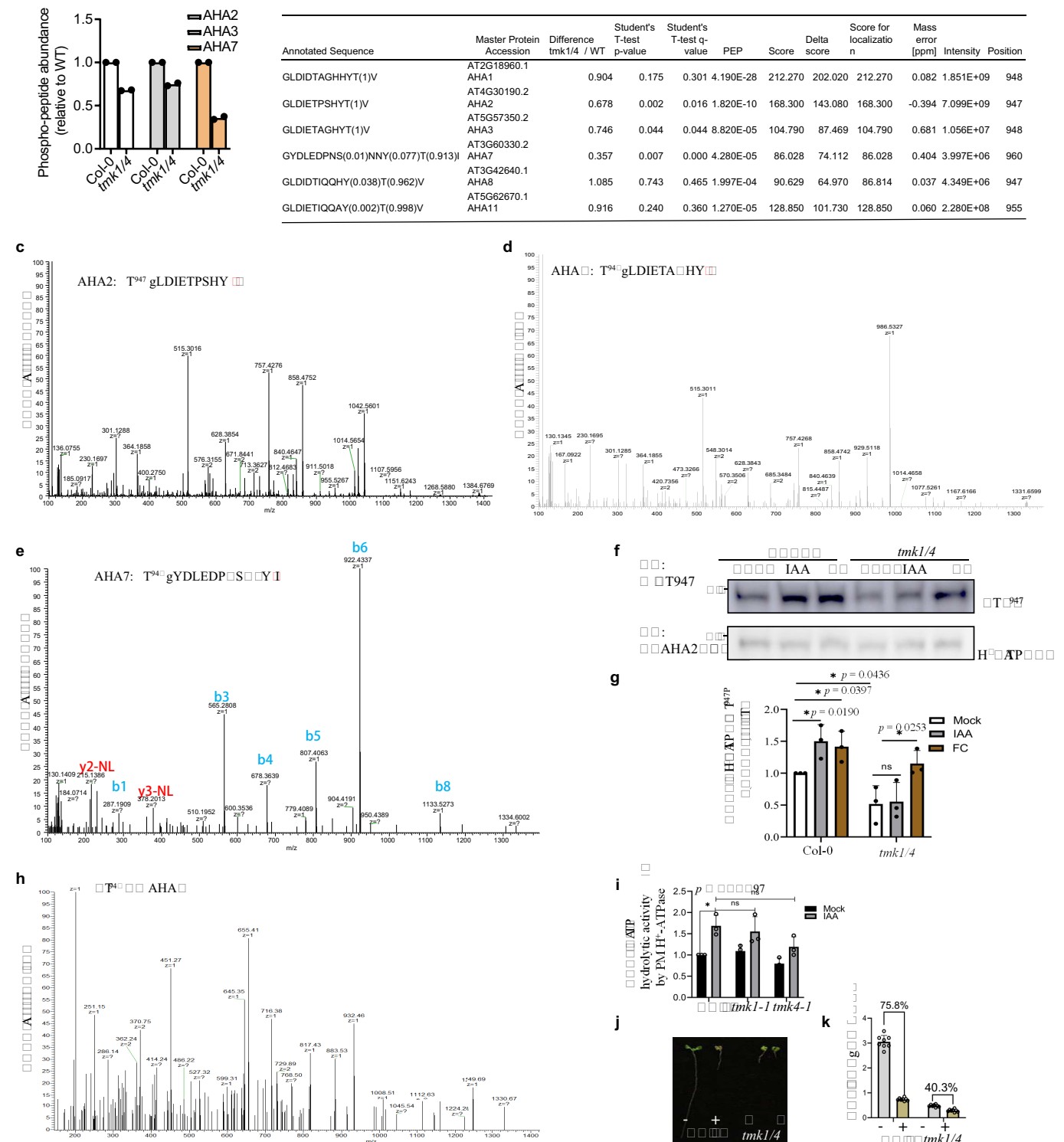

**Extended Data Fig. 2** | See next page for caption.

**Extended Data Fig. 2 | TMK1 and TMK4 impact the phosphorylation status of AHAs and the function of the PM H⁺-ATPase pump. a**, The phosphorylation status of AHAs was changed in the *tmk1-1 tmk4-1* (*tmk1/4)* mutant. The aerial part of 5-days auxin-depleted seedlings was used to prepare membrane proteins for TMT (Tandem mass tag) labelling and mass spectrometry quantification as described in Method. Mass spectrometry analysis showed that the abundance of the peptides containing phosphorylated penultimate threonine from AHA2, AHA3, and AHA7 was significantly decreased in *tmk1-1 tmk4-1* (*tmk1/4*) mutant relative to wild type. Values are means; n = 2. **b**, Summary of phosphorylated peptides mass spectrometry information. The C-terminal peptides of AHA1, AHA2, AHA3, AHA7, and AHA11 containing phosphorylated penultimate threonine were identified from mass spectrometry analysis. **c-e**, High-resolution fragmentation spectra of AHA2 (**c**), AHA3(**d**), and AHA11(e) C-terminal peptides containing phosphorylated penultimate threonine are presented. **f**, The *tmk1-1 tmk4-1*(*tmk1/4*) mutant is insensitive to auxin inducing AHA phosphorylation but remains sensitive to fusicoccin. The endogenous auxin-depleted aerial sections of seedlings were incubated with 100 nM IAA for 10 min or 10 μm fusicoccin (FC) for 5 min, respectively. The amounts of AHA proteins and the phosphorylation status of the penultimate Thr in the C terminus were determined by immunoblot analysis with anti-AHA (H⁺-ATPase) and anti-pThr-947 (pThr 947) antibodies, respectively. **g**, Quantification of the phosphorylation level of the H⁺-ATPase. Values are means ± SD; n = 3 independent biological replicates, *P ≤ 0.05; ns, no significant, results of two way ANOVA test. **h**, Fragmentation spectra of peptides containing phosphorylated penultimate threonine of AHA1-C16 synthetical peptide (pT948 of AHA1) (see Fig. 2e). **i**, Auxin induction of H⁺-ATPase activity in the aerial parts of wild type, the *tmk1-1*, and *tmk4-1* mutant. Aerial sections of 14-days old seedlings were treated with 10 μm IAA for 30 min and used for vanadate-sensitive ATP hydrolysis assay by determining the inorganic phosphate released from ATP as described previously[27]. The values shown are relative ATP hydrolytic activity of indicated samples to that of control Col-0 without auxin treatment. Values are means ± SD; n = 3. *P ≤ 0.05; ns, not significant. The results were analysed by a two-way ANOVA test. **j**, Lithium tolerance in the *tmk1-1 tmk4-1* (*tmk1/4*) mutant. Wild type (Col-0) and *tmk1-1 tmk4-1* (*tmk1/4*) mutant seedlings were grown on 1/2 MS medium with or without 18 mM LiCl for 5 days. LiCl treatment caused severe seedling growth retardation and severe chlorosis of the aerial parts in Col-0, whereas the *tmk1-1 tmk4-1* (*tmk1/4*) mutant was tolerant to LiCl, especially in the aerial parts. **k**, The root length of the seedlings was measured by ImageJ. Values are means ± SD, n = 8. The number above the columns indicates the percentage of root growth inhibition induced by LiCl.

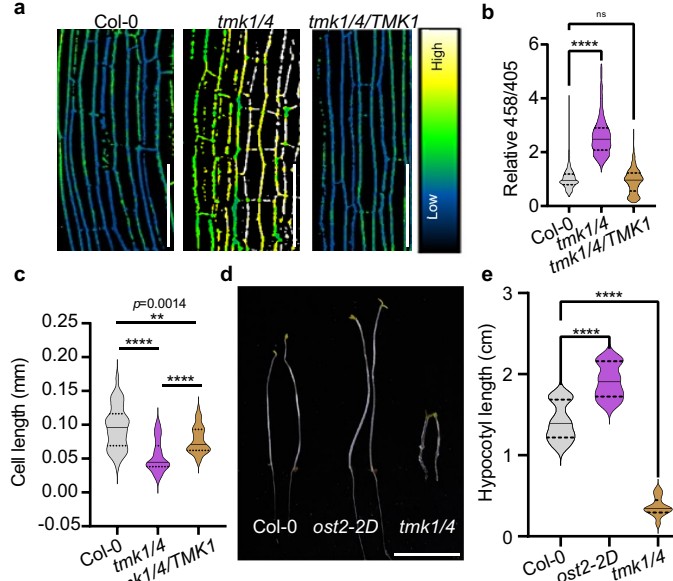

**Extended Data Fig. 3 | TMK1 and TMK4 regulate apoplastic pH and hypocotyl elongation. a**, TMK1 (*pTMK1::TMK1-GFP*) restored the apoplastic pH changes in the *tmk1-1 tmk4-1* (*tmk1/4*) mutant. Comparison of the apoplastic pH in WT, the *tmk1-1 tmk4-1* mutant, and the *tmk1-1 tmk4-1/TMK1-GFP* (*tmk1/4/TMK1*) complemented line. Visualized by HPTS staining (**a**). Y-Axis: the mean 458/405 values of the *tmk1-1 tmk4-1* mutant and the *TMK1* complemented line relative to wild type (**b**). 3 independent assays were conducted with similar results. **c**, Epidermal cell lengths of hypocotyls from two days-old etiolated seedlings were measured using Image J. Hypocotyl epidermal cells in the 100-500 µM region after apical hook were measured. n = 41, 52, and 53 for Col-0, *tmk1/4* and *tmk1/4/TMK1*. The results were analysed by one Way ANOVA tests in b, c. ****P≤0.0001. **d** and **e**, The *tmk1-1 tmk4-1* mutant showed a defect in hypocotyl elongation (**c**). Hypocotyl lengths of 3 days-old etiolated seedlings were measured by Image J (**d**). n = 21, 11, and 14 for Col-0, *ost2-2D*, and *tmk1/4*, respectively. The results were analysed by one-way ANOVA tests in b, c, e. Scale bar = 100 µM (**a**), or 1 cm (**d**). ** P ≤ 0.01, **** P ≤ 0.0001; ns, not significant.

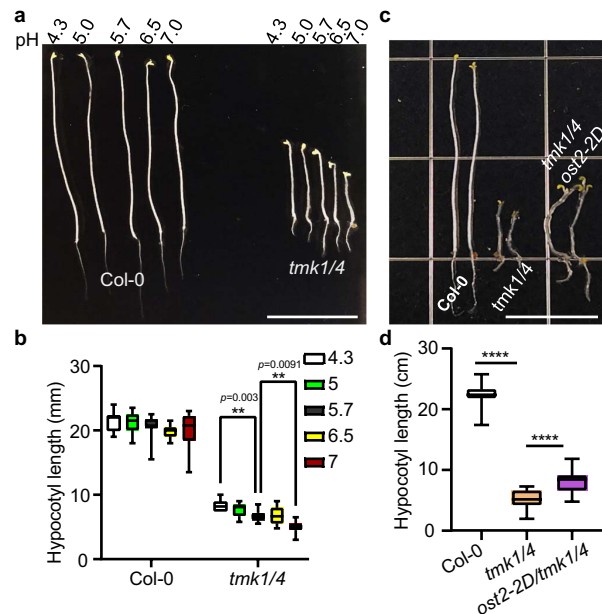

**Extended Data. Fig. 4 | Acidic environments and activation of the PM H⁺-ATPase pump partially restored hypocotyl elongation defect in *tmk1-1 tmk4-1*.** a and b, Low pH in the medium was able to partially restore the *tmk1-1 tmk4-1* (*tmk1/4*) defect in hypocotyl elongation. Seedlings were grown on 1/2 MS medium with indicated pH (**a**), and hypocotyl lengths were measured by ImageJ (**b**). The box plot centre lines represent median; box limits indicate the 25th and 75th percentiles; and whisker extend 1.5xIQR from the 25th and 75th percentiles (n = 15 hypocotyls per line). c and d, *ost2-2D* mutation partially restored the hypocotyl elongation defect of *tmk1-1 tmk4-1* mutant. Seedlings were grown on a 1/2 MS medium for 4 days, and hypocotyl lengths were measured by ImageJ. The box plot centre lines represent median; box limits indicate the 25th and 75th percentiles; and whisker extend 1.5xIQR from the 25th and 75th percentiles (*n* = 36 for Col-0, n = 24 for *tmk1/4* and n = 21 for *ost2-2D/tmk1/4*). Results were analysed by one-way ANOVA tests in b, d. Scale bar = 10 mm. ** $p \le 0.01$, **** $p \le 0.0001$.

# Reporting Summary

## Statistics

For all statistical analyses, confirm that the following items are present in the figure legend, table legend, main text, or Methods section.

| n/a | Confirmed | |
|---|---|---|
| ☐ | ☒ | The exact sample size ($n$) for each experimental group/condition, given as a discrete number and unit of measurement |
| ☐ | ☒ | A statement on whether measurements were taken from distinct samples or whether the same sample was measured repeatedly |
| ☐ | ☒ | The statistical test(s) used AND whether they are one- or two-sided<br>*Only common tests should be described solely by name; describe more complex techniques in the Methods section.* |
| ☒ | ☐ | A description of all covariates tested |
| ☐ | ☒ | A description of any assumptions or corrections, such as tests of normality and adjustment for multiple comparisons |
| ☐ | ☒ | A full description of the statistical parameters including central tendency (e.g. means) or other basic estimates (e.g. regression coefficient) AND variation (e.g. standard deviation) or associated estimates of uncertainty (e.g. confidence intervals) |
| ☐ | ☒ | For null hypothesis testing, the test statistic (e.g. $F$, $t$, $r$) with confidence intervals, effect sizes, degrees of freedom and $P$ value noted<br>*Give P values as exact values whenever suitable.* |
| ☒ | ☐ | For Bayesian analysis, information on the choice of priors and Markov chain Monte Carlo settings |
| ☒ | ☐ | For hierarchical and complex designs, identification of the appropriate level for tests and full reporting of outcomes |
| ☒ | ☐ | Estimates of effect sizes (e.g. Cohen's $d$, Pearson's $r$), indicating how they were calculated |

*Our web collection on statistics for biologists contains articles on many of the points above.*

## Software and code

Policy information about availability of computer code

| | |
|---|---|
| Data collection | Proteomic data was collected by Orbitrap Fusion (Thermo Fisher Scientific, Watham, MA), by software Thermo Xcalibur 3.0.63 |
| Data analysis | 1: Arabidopsis root and hypocotyl tissue and cell length are all measured by ImageJ (JAVA 1.8.0_172). 2: FRET analyzer , an ImagJ plug-in, was used to analyze FRET signal. 3: The ratio metric image was analyzed and quantified by Fiji (JAVA 1.8.0_172), using a macro language, which was described in the manuscript. 4:LC-MS data were analyzed with Maxquant (version 1.6.2.6) with Andromeda search engine. |

For manuscripts utilizing custom algorithms or software that are central to the research but not yet described in published literature, software must be made available to editors and reviewers. We strongly encourage code deposition in a community repository (e.g. GitHub). See the Nature Portfolio guidelines for submitting code & software for further information.

## Data

Policy information about availability of data

All manuscripts must include a data availability statement. This statement should provide the following information, where applicable:

- Accession codes, unique identifiers, or web links for publicly available datasets
- A description of any restrictions on data availability
- For clinical datasets or third party data, please ensure that the statement adheres to our policy

Code and Data availability statement was included in Methods Section.  Mass spectrometry raw data associated with Figure 2e, extended Data Figure 2a,b are available at the MassIVE under accession number: MSV000087822.Source Data (gel and graphs) are provided with manuscript. The data supporting the findings in this study are available and described  within the manuscript and extend data information file.

# Field-specific reporting

Please select the one below that is the best fit for your research. If you are not sure, read the appropriate sections before making your selection.

☒ Life sciences ☐ Behavioural & social sciences ☐ Ecological, evolutionary & environmental sciences

For a reference copy of the document with all sections, see nature.com/documents/nr-reporting-summary-flat.pdf

# Life sciences study design

All studies must disclose on these points even when the disclosure is negative.

| | |
|---|---|
| Sample size | Sample size calculation was not performed. We determined the number of samples in each experiment as commonly accepted standards in the field. 1: Western blot. All the western blot assays were repeat 3-4 times and the intensity peak of the target bands were measured by ImageJ. 2: Quantitative analysis of hypocotyl elongation zone apoplastic pH value. Each data set was from the measurement of 10-20 etiolated hypocotyls of different genotypes. 3: MS to identified in vitro peptides phosphorylation. Two biological repeats were included, in each biological repeat, 3 independent technique repeats were included. 4: TMT-label based phosphoproteomics MS. 0.45 mg proteins of each sample were applied to TMT-labeled and ms analysis. 2 independent biological repeats were performed for this assay. 1-5 mg proteins were applied to IP-MS. 5: For FRET assay. The data was collected and analyzed from 10 individual cells. The experiment was repeated 3 times. |
| Data exclusions | No data were excluded from this study. |
| Replication | The data in thsi paper is highly replicable, as the companion paper conduct several experiments independently used same materials produced same results. All the measures in this study were conducted in 2-4 times biology repeats, which start from germination of the seedlings. Each set of the data were collected and analyzed independently. |
| Randomization | The study does not involved work that required random allocation. The sample were allocated into experimental groups based on their genotypes, for instance, by wild type or specific gene mutations. The randomization was not applied in this study. |
| Blinding | No double blinding is applied in this study. For this current study, blinding is not relevant. |

# Reporting for specific materials, systems and methods

We require information from authors about some types of materials, experimental systems and methods used in many studies. Here, indicate whether each material, system or method listed is relevant to your study. If you are not sure if a list item applies to your research, read the appropriate section before selecting a response.

## Materials & experimental systems

| n/a | Involved in the study |
|---|---|
| ☐ | ☒ Antibodies |
| ☒ | ☐ Eukaryotic cell lines |
| ☒ | ☐ Palaeontology and archaeology |
| ☒ | ☐ Animals and other organisms |
| ☒ | ☐ Human research participants |
| ☒ | ☐ Clinical data |
| ☒ | ☐ Dual use research of concern |

## Methods

| n/a | Involved in the study |
|---|---|
| ☒ | ☐ ChIP-seq |
| ☒ | ☐ Flow cytometry |
| ☒ | ☐ MRI-based neuroimaging |

## Antibodies

| | |
|---|---|
| Antibodies used | 1: The anti-HA (Invitrogen, # 26183, 1:2000 dilution), GFP (Chromotek, #3h9, 1:1000 dilution) , Myc (sinobiological, #100029-MM08, 1:1000 dilution), GST (Santa Cruz, #sc-138, 1:1000 dilution) and MBP (Invitrogen, PA1-989, 1:1000 dilution ) antibodies that were used in this study are all commercial available with the validations.<br>2: pT947 AHA antibody was described in manuscript, which was generated from rabbit (1:5000 dilution). |
| Validation | Validation statements of commercial primary antibodies are available from manufacturers. α-GFP (https://www.chromotek.com/fileadmin/content/PDFs/Data_Sheets/3h9_Datasheet_GFP_antibody__3H9.pdf), α-HA-HRP (https://www.thermofisher.com/order/genome-database/dataSheetPdf producttype=antibody&productsubtype=antibody_primary&productId=26183-HRP&version=133), α-myc (http://www.sinobiologicalcdn.com/reagent/100029-MM08.pdf). α-MBP (https://www.thermofisher.com/order/genome-database/dataSheetPdf?producttype=antibody&productsubtype=antibody_primary&productId=PA1-989&version=133).α-GST (https://datasheets.scbt.com/sc-138.pdf). pT947 antibody was validated as reference: doi:10.1093/pcp/pcq078. |

