## [Peer Review File · Nature]

Manuscript Title: TMK-based cell surface auxin signaling activates cell wall acidification in Arabidopsis

Reviewer Comments & Author Rebuttals

Reviewer Reports on the Initial Version:

Referee #1 (Remarks to the Author):

In the manuscript "TMK-based cell surface auxin signaling activates cell wall acidification in Arabidopsis" Wenwei Li and colleagues investigate the molecular mechanisms by which auxin leads to cell wall acidification and subsequently alters cellular expansion. They identify an interaction of the receptor kinase TMK1, involved in auxin perception, and the plasma membrane H⁺ - ATPase AHA. This interaction leads to AHA phosphorylation, acidification of the cell wall in response to auxin, and regulates cellular elongation. Overall, this is good work and resembles a step in our understanding of how auxin regulates cellular elongation. The conclusions are largely warranted by the presented data. While the manuscript is well written in a concise manner, the quality of the figures and their formatting is not exemplary. There are several issues that I'd like to highlight.

Major:

1. Not all figures seem to contain statistical analyses; E.g. no error bars and significance tests in Ext. Data Figure 2a; Statistics are missing for Fig 2a,b where it is also unclear how many independent experiments were performed; data presented in Fig. 2e would require 2-way ANOVA for proper analysis ; Stats seem missing in Ext. Data Fig 3d
2. The lithium resistance data doesn't add much as no auxin involvement was tested and because the physiological difference of the mutant to WT might lead to numerous consequences that are not related to acidification and/or auxin.

Minor:

1. L58: TMK1 has been also implicated in temperature dependent root growth regulation (Gaillochet et al, 2020, Development)
2. Extended Data Figure 1 is of low quality and the labelling seems off.
3. It is unclear how many biological repeats were used for the FRET measurements
4. Fig. 1d show co-IP with spectacular time resolution but it is not described how this rapid treatment was conducted.
5. Extended Data Figure 2 is largely unreadable.
6. L296-299: Sentence is unclear.
7. L309: Perez et al. manuscript in preparation; as the data is not available yet it doesn't seem to make sense to reference it

Referee #2 (Remarks to the Author):

Plasma membrane proton ATPases of plants are central to metabolism and growth, and are known to be activated by phosphorylation. This manuscript identifies a family of receptor kinases (TMKs) that can phosphorylate H⁺-ATPases at a critical residue near the C-terminus, and reports that the kinases are needed to acidify the extracellular space. They also provide evidence that the growth hormone auxin signals through the TMK kinases. Together with previous work showing that these kinases affect auxin responses, the results suggest that they are key players in the signaling that regulates growth in response to auxin. The interpretations are complicated by the presence of a

second auxin-regulated pathway that is also required for growth, and that acts through nuclear auxin receptors and changes in transcription. Also, it seems likely that additional signals may regulate the TMKs or other kinases that feed into the system. I suspect that some aspects of the results have been affected by these parallel pathways. Taking into account these difficulties, the work represents an important step forward in our understanding of plant growth control. In particular, it adds to growing evidence that auxin controls growth through multiple signaling pathways that act through distinct classes of receptors. In this respect auxin response now seems analogous to cases in animals where quite different pathways can respond to the same signal (such as steroid hormones acting through both cytoplasmic/nuclear and membrane receptors, or Wnt regulating both canonical and non-canonical pathways). The work also reveals a class of kinases that can activate the proton pump, potentially one of several to be discovered, given the variety of circumstances in which the pump is activated.

In Figure 1 they provide several lines of evidence that TMKs can interact with H⁺-ATPases, including IP-Mass spectrometry, co-IP from transgenic plants expressing fusion proteins or from transiently transfected protoplasts, and FRET in transiently transfected cells. In their initial IP-mass spectrometry with TMK-GUS protein, they should provide a supplementary dataset indicating what proteins they isolated in addition to H⁺-ATPases. The FRET assays have not been quantitated, and this would be helpful. The auxin-induced association is modest (Fig. 1d). I wonder whether they could buttress this result by auxin treatments in the FRET assays. Does the auxin-induced association depend on the TMK kinase activity?

Figure 2 provides evidence that TMKs can phosphorylate H⁺-ATPases at the activating Thr947 residue. The *in vitro* data (Fig. 2c,d) are more clear than the *in vivo* data in Fig. 2a,b,e. In Figure 2a,b, auxin increases PhosphoThr947 by only about 35% in wild type, and the relative effect looks similar in the *tmk1 tmk4* double mutant. The much larger effect is on the baseline phosphoThr947 level, which is about 20% of wild type in the *tmk1 tmk4* double mutant. It would seem appropriate to discuss whether the baseline in wild type arises from residual or prior auxin response, or whether it may represent a baseline activity of TMKs in the absence of auxin signaling or in response to other signals. In Fig. 2e, the *tmk1 tmk4* mutant has clearly lost the auxin stimulation of H⁺-ATPase activity, but in this assay the mutant baseline activity is similar to wild type. Does this mean that the ATPase activity does not depend solely on Thr947 phosphorylation (hence differing from Figure 2a,b), or does the ATPase activity assay also measure other proteins? Both the anti-phosphoThr947 assay and the ATPase assay were done in "auxin-depleted" tissue, but it is not clear whether the same protocol was used, and the auxin level and treatment time were different. It could be helpful to use the same conditions for the assays in fig. 2a,b vs. Fig. 2e, to allow easier comparison. Also, why not test how fusicoccin affects phosphorylation in the *tmk1 tmk4* plants, although this is less important than the auxin stimulation. Finally, I wonder whether they can exclude the model that TMKs phosphorylate H⁺-ATPases independently of auxin, and that the apparent *in vivo* auxin effect is actually due to effects of SAUR induction at the 10 or 30 minute time points they have assayed.

Figure 2b should include statistical analysis.

Figure 4 presents evidence using a fluorescent pH indicator that *tmk1 tmk4* mutants have higher apoplastic (extracellular space) pH, consistent with the effects on proton pump phosphorylation. I find that Figure 3 does not add much that is not also in Figure 4, so these two figures could be combined, and all or part of the data in Figure 3 put in a supplementary figure. The pH changes with and without auxin in wild type and mutant are the most important aspect here, and the shorter hypocotyl and shorter cells of the *tmk1 tmk4* mutant have been reported before anyway.

Fig. 4c. It would be interesting to test growth stimulation of *tmk1 tmk4* mutant hypocotyls in response to auxin in whole seedling assays, not just the short-term assay with excised hypocotyl segments, which only gives a response of about 10%. Whole-seedling assays work best in light-grown seedlings using a synthetic auxin such as picloram.

Cao et al. reported previously that auxin induces cleavage of TMK1 to affect nuclear auxin responses. It would seem relevant to discuss how the cleavage might affect the H⁺-ATPase activation. Does the cleavage activate the portion of the kinase that is still at the PM, or perhaps inactivate it to then limit H⁺-ATPase activation?

lines 147-150. Why would expression in *E. coli* cause phosphorylation of Thr947? If anything I would have expected the immunoprecipitated H⁺-ATPase to be phosphorylated already. Perhaps the *E. coli*-expressed protein folds incorrectly.

They present the apoplastic pH data as relative fluorescence at two different wavelengths. Can the degree of pH change be estimated, for example by calibrating the dye with buffers?

Referee #3 (Remarks to the Author):

Auxin mediated cell expansion has been the subject of intense research for decades and the acid growth hypothesis has been central to helping explain the rapid (i.e. minutes) non-genomic stimulus of growth. However, the molecular mechanism(s) regulating auxin mediated cell expansion remains poorly understood.

The current manuscript by Lin et al throws new light on this previously intractable problem through their characterization of the PM-localized TMK receptor-like kinases (and its interactive partners/targets). Using an IP-MS based approach, Lin et al show that TMK receptor kinases interact with members of AHA gene family encoding PM ATPases. They further show that TMK kinase domain (but not the extracellular domain) interacts with AHAs in an auxin dependent fashion using an elegant FRET based approach. It is known that the phosphorylation of the conserved penultimate threonine (Thr) residue in the AHA family members is crucial for the activation of these PM-ATPases. Lin et al show that auxin activates AHA phosphorylation through TMKs.

Phosphorylation of the penultimate Thr is known to be crucial for the activation of AHAs. Pull down assays using anti-pThr-947 antibody recognizing penultimate Thr, Lin et al show that the level of phosphorylated penultimate Thr was reduced in the *tmk1/4* mutant. Moreover, the activation of PM H⁺-ATPase is coupled with ATP hydrolysis and this was abolished in the *tmk1/4* mutant. Uptake of lithium ions is coupled with the activation of PM ATPase and in a very elegant experiment, Lin et al show that *tmk1/4* mutant seedlings were more tolerant to toxic lithium ions than the wild type. The authors then show that auxin mediated apoplast acidification is correlated with increase in hypocotyl cell length. This auxin mediated apoplastic acidification and increase in hypocotyl cell length is abolished in *tmk1/4* mutants but is restored when complemented with TMK1. This provides a direct mechanism for acidification of apoplast and cell expansion.

The research is of high quality and is of general interest to plant biologists. The manuscript is well-written, and experiments are logical and well designed. There are some issues with the manuscript that need to be resolved to further improve the quality of the manuscript.

One of the key concerns is that the auxin dependent nature of AHA-TMK interaction is performed in protoplasts. Given that protoplasts don't have cell walls and the nature of this experiment in interpreting this is as an auxin dependent acid-cell response, should this not be done in a system with a cell wall? How realistic are the conditions on the plasma membrane of a protoplast? What type of cells are these? Have they de-differentiated?

Despite this concern, the experiments presented in figures 3 and 4 where the authors show that

auxin mediated apoplast acidification is correlated with increase in hypocotyl cell length and this auxin mediated acidification of apoplast and increase in hypocotyl cell length is abolished in *tmk1/4* mutants but rescues by TMK1 is convincing and provides a direct mechanism for apoplastic acidification and cell expansion.

It is clear that penultimate Thr is crucial for the function of AHAs and authors have shown that pull down assays using anti-pThr-947 antibody that the level of phosphorylated penultimate Thr was reduced in the *tmk1/4* mutant. It would be interesting to check the hypocotyl cell elongation upon auxin treatment in *tmk1/4* mutants complemented with kinase dead TMK1.

The phosphorylation experiments presented in figure 2 are generally well described but it is not clear which tissues have been used for these assays. Though this information is present in the figure legend, it will be useful to mention this in the main text too.

In some of the experiments, the authors have used non physiological concentration of NAA (fig1 1uM NAA). This is not ideal and it will be useful to repeat the assays using more physiological concentrations.

Pull down assays using pTMK1::TMK1-GFP lines are very nice (fig1 A) but am puzzled why the authors decided to use 35S::TMK1:GFP plants (Fig 1B) when they also had the opportunity to use pTMK1::TMK1-GFP lines that provides more spatial resolution. Also, whilst I understand the authors were interested in AHAs, there is value in also reporting what other proteins came out in these TMK immunoprecipitation assay.

The experiments with toxic lithium ions (extended figure 3) and fusicoccin are very elegant. I suggest authors consider moving the Lithium data from extended figure 2 to the main figure 2.

The authors also need to provide more details in the methods section. For example, it is not clear which tissues have been used for Co-IP studies (eg whole plants, leaves??). Also, there are no methods provided for FRET analysis. Information on what statistical tests have been done is often vague and it would be useful that this information is provided in the figure legends.

Author Rebuttals to Initial Comments:

Point-by-Point Responses to Referees

Summary of responses: We thank all three referees for your overall positive comments on our work and your constructive critiques, which have certainly helped to improve our manuscript. Please note that we have addressed the concerns that require additional experimentation or data analyses and explained our preferences for certain styles of data presentation or figure arrangement, but we will be flexible on these formatting issues. Please note that changes/additions in the main text are highlighted in yellow. Please see below for our point-by-point responses in blue.

Referees' comments:

Referee #1 (Remarks to the Author):

In the manuscript "TMK-based cell surface auxin signaling activates cell wall acidification in Arabidopsis" Wenwei Li and colleagues investigate the molecular mechanisms by which

auxin leads to cell wall acidification and subsequently alters cellular expansion. They identify an interaction of the receptor kinase TMK1, involved in auxin perception, and the plasma membrane H⁺ - ATPase AHA. This interaction leads to AHA phosphorylation, acidification of the cell wall in response to auxin, and regulates cellular elongation. Overall, this is good work and resembles a step in our understanding of how auxin regulates cellular elongation. The conclusions are largely warranted by the presented data. While the manuscript is well written in a concise manner, the quality of the figures and their formatting is not exemplary. There are several issues that I'd like to highlight.

Major:

1. Not all figures seem to contain statistical analyses; E.g. no error bars and significance tests in Ext. Data Figure 2a; Statistics are missing for Fig 2a,b where it is also unclear how many independent experiments were performed; data presented in Fig. 2e would require 2-way ANOVA for proper analysis ; Stats seem missing in Ext. Data Fig 3d

Response: We thank the referee for these constructive critiques. We have addressed these concerns as outlined below:

- 1) Error bars and significance tests have been added to Ext Data Fig 2a. The information on the statistical analysis is presented in the summary table, including p and q-values.
- 2) Statistical analysis is now included in Fig. 2a,b.
- 3) Data in Fig 2e has been re-analyzed using 2-way ANOVA with more replicates.
- 4) Statistical analysis is included in Ext Data Fig. 3d.

2. The lithium resistance data doesn't add much as no auxin involvement was tested and because the physiological difference of the mutant to WT might lead to numerous consequences that are not related to acidification and/or auxin.

Response: We agree with this referee that the lithium resistance data by itself doesn't add much. However, the data is consistent with a role for TMK1/4 in the regulation of apoplastic pH. Importantly this role is directly supported by a series of other data described in our manuscript. Furthermore, referee #3 liked this data and requested that this data be moved to the main figure (see below). Therefore, we incline to leave this data in Ext Data Fig. 2h, i.

Minor:

1. L58: TMK1 has been also implicated in temperature dependent root growth regulation (Gaillochet et al, 2020, Development)

Response: The reference has been cited

2. Extended Data Figure 1 is of low quality and the labelling seems off.

Response: We replaced this data with a high-quality figure with correct labeling.

3. *It is unclear how many biological repeats were used for the FRET measurements.*

Response: The FRET analysis in tobacco leaves is now replaced with a FRET analysis in Arabidopsis protoplasts to show a rapid induction of TMK1-AHA1 interaction by auxin treatment. The number of biological repeats and statistical analysis for this new data are shown in Fig, 1g.

4. *Fig. 1d show co-IP with spectacular time resolution but it is not described how this rapid treatment was conducted.*

Response: The detailed procedures of this assay were described in Materials and Methods. Briefly, the samples were frozen in liquid N₂ immediately after a period of treatment to stop the reaction and then subjected to further Co-IP assay.

5. *Extended Data Figure 2 is largely unreadable.*

Response: We re-organized this Figure and used images with high resolution.

6. *L296-299: Sentence is unclear.*

Response: We revised this sentence as suggested. It now reads "*These data suggest that upon activation by auxin, the cell surface auxin signaling component TMKs act as a protein kinase to directly and rapidly initiate the phosphorylation and activation of PM H⁺-ATPase, although our findings do not exclude the possibility that the TMK1-mediated AHA phosphorylation may also respond to other stimuli*" (see lines 334-339).

7. *L309: Perez et al. manuscript in preparation; as the data is not available yet it doesn't seem to make sense to reference it*

Response: we have removed this information.

Referee #2 (Remarks to the Author):

Plasma membrane proton ATPases of plants are central to metabolism and growth, and are known to be activated by phosphorylation. This manuscript identifies a family of receptor kinases (TMKs) that can phosphorylate H⁺-ATPases at a critical residue near the C-terminus, and reports that the kinases are needed to acidify the extracellular space. They also provide evidence that the growth hormone auxin signals through the TMK kinases. Together with previous work showing that these kinases affect auxin responses, the results suggest that they are key players in the signaling that regulates growth in response to auxin. The interpretations are complicated by the presence of a second auxin-regulated pathway that is also required for growth, and that acts through nuclear auxin receptors and changes in transcription. Also, it seems likely that additional signals may regulate the TMKs or other kinases that feed into the system. I suspect that some aspects of the results have been affected by these parallel pathways. Taking into account these difficulties, the work represents an important step forward in our understanding of plant growth control. In particular, it adds to growing evidence that auxin controls growth through multiple signaling pathways that act through distinct classes of receptors. In this respect auxin response now seems analogous to cases in animals where quite different pathways can respond to the same signal (such as steroid hormones acting through both cytoplasmic/nuclear and membrane receptors, or Wnt regulating both canonical and non-canonical pathways). The work also reveals a class of kinases that can activate the proton pump, potentially one of several to be discovered, given the variety of circumstances in which the pump is activated.

We thank this referee for the positive comment on the significance of our work.

In Figure 1 they provide several lines of evidence that TMKs can interact with H⁺-ATPases, including IP-Mass spectrometry, co-IP from transgenic plants expressing fusion proteins or from transiently transfected protoplasts, and FRET in transiently transfected cells. In their initial IP-mass spectrometry with TMK-GUS protein, they should provide a supplementary dataset indicating what proteins they isolated in addition to H⁺-ATPases. The FRET assays have not been quantitated, and this would be helpful. The auxin-induced association is modest (Fig. 1d). I wonder whether they could buttress this result by auxin treatments in the FRET assays. Does the auxin-induced association depend on the TMK kinase activity?

Response: We appreciate the referee's excellent suggestion to add new data on auxin promotion of TMK-AHA interaction revealed by FRET assay. We have now replaced the FRET assay in tobacco leaves with a new FRET assay in Arabidopsis protoplasts involving a microfluidic setup that enables a time-course analysis of TMK1-AHA1 interaction with high temporal resolution. Our new results show that auxin induces the interaction within seconds. This interesting new data is shown in Fig. 1e-g.

We agree with this reference that some IP-MS data may be of interest to the readers, but we also did not want to mislead the readers, as many proteins identified from IP-MS analysis may be false positives, but require testing with other independent methods as we described here for AHAs. We feel that non-validated candidate proteins should not be published to

avoid potentially misleading information. Furthermore, our story here focuses on the role of TMKs in the regulation of AHAs, and other identified proteins will not add anything to this story. Therefore, we incline not to include the entire IP-MS data in this manuscript at this point. To be flexible, we would be happy to include the data if the referee and the editor feel strongly otherwise.

Figure 2 provides evidence that TMKs can phosphorylate H⁺-ATPases at the activating Thr947 residue. The in vitro data (Fig. 2c,d) are more clear than the in vivo data in Fig. 2a,b,e. In Figure 2a,b, auxin increases PhosphoThr947 by only about 35% in wild type, and the relative effect looks similar in the tmk1 tmk4 double mutant. The much larger effect is on the baseline phosphoThr947 level, which is about 20% of wild type in the tmk1 tmk4 double mutant. It would seem appropriate to discuss whether the baseline in wild type arises from residual or prior auxin response, or whether it may represent a baseline activity of TMKs in the absence of auxin signaling or in response to other signals. In Fig. 2e, the tmk1 tmk4 mutant has clearly lost the auxin stimulation of H⁺-ATPase activity, but in this assay the mutant baseline activity is similar to wild type. Does this mean that the ATPase activity does not depend solely on Thr947 phosphorylation (hence differing from Figure 2a,b), or does the ATPase activity assay also measure other proteins? Both the anti-phosphoThr947 assay and the ATPase assay were done in "auxin-depleted" tissue, but it is not clear whether the same protocol was used, and the auxin level and treatment time were different. It could be helpful to use the same conditions for the assays in fig. 2a,b vs. Fig. 2e, to allow easier comparison. Also, why not test how fusicoccin affects phosphorylation in the tmk1 tmk4 plants, although this is less important than the auxin stimulation. Finally, I wonder whether they can exclude the model that TMKs phosphorylate H⁺-ATPases independently of auxin, and that the apparent in vivo auxin effect is actually due to effects of SAUR induction at the 10 or 30 minute time points they have assayed.

Response: We greatly appreciate these very constructive comments and have addressed them as outlined below:

1) In the original manuscript, we examined the effect of auxin treatment on the level of phosphorylated T947 by treating seedlings with 100 nM IAA for 10 minutes. To allow for the comparison of T947 phosphorylation with ATPase hydrolysis, as the referee suggested, we examined the effect of exogenous IAA on T947 phosphorylation using the identical conditions that were used for the ATPase hydrolysis assay (Fig.2e), i.e., treatment with 10 μ M IAA for 30 minutes. Under this condition, we observed nearly an 80 % increase in the phosphorylated T947 in WT, but only a small increase (about 10%) in the *tmk1/4* mutant, which is not statistically significant, when compared to the mock treatment (Fig. 2a, b). We also repeated the experiments with 100 nM IAA treatments. After including additional replicates, we observed similar trends as with 10 μ M IAA, although the effect in the wild type was smaller (Ext data Fig. 2d,e).

2) Under the condition described above, we found that the basal level of phosphorylated T947 in the *tmk1/4* mutant is about 50% of the wild-type level, which is consistent with our TMT-label phosphoproteomic survey (Ext Data, Fig. 2a,b). The MS survey revealed that among several identified AHAs, the phosphorylated level of penultimate Thr residue was reduced by 10-65%, respectively. In various experiments, we often observed large variations in the wild type baseline phosphorylation levels (see Fig. 2b; Ext data Fig. 2d,e). These variations may reflect the sensitivity of AHA phosphorylation to various stimuli, as light and other hormones. It is possible that at least some other stimuli might affect AHA phosphorylation via TMKs directly or indirectly, which could explain the larger effect of *tmk1/4* on baseline phosphorylation observed sometimes.

3) ATPase hydrolysis activity in both wild type and *tmk1/4* is variable among experiments as observed for T947 phosphorylation levels. We repeated the experiments and included more biological replicates in our quantitative analysis, and found that there is a significant difference in the baseline level ATPase activity between wild type and *tmk1/4* (Fig. 2e).

4) As described above, we repeated the phosphorylation experiments using identical conditions used for ATP hydrolysis assay and found that the results were similar to those from our previous experiments involving lower auxin levels for phosphorylation assay (See Fig. 2a,b).

5) As suggested by the referee, we tested the effect of fusicoccin (FC) on the T947 phosphorylation in *tmk1/4*, and the result is now included Ext Data Fig. 2d,e. FC is known to promote the association of 14-3-3 with the C-terminal region of AHA containing the phosphorylated penultimate Thr, resulting in the de-repression of C terminal autoinhibition to the AHA catalytic activity. Our result showed that FC increased the level of phosphorylation T947 in the *tmk1/4* mutant, although the induction level is lower than that in WT. Thus the PM-H⁺ATPase in *tmk1/4* is capable of being phosphorylated in response to other stimuli.

6) Is TMK phosphorylation of AHA independent of auxin or linked to SAUR?

Our data clearly demonstrated that auxin phosphorylates AHA at least in part via TMKs. However, as the referee hinted we cannot exclude the possibility that TMK might phosphorylate AHAs independent of auxin. A sentence discussing this possibility is included in the main text (page 14, lines #339-346).

Because auxin-induced association of TMKs with AHA is very rapid (occurring in seconds) and because TMKs directly phosphorylate AHAs, we hypothesize that auxin-activated TMKs directly initiate AHA phosphorylation but not via changes in SAUR expression, while SAUR-

mediated inhibition of PP2Cs is to sustain AHA phosphorylation. As described above, our new results showed that 10 μ M auxin induced an 80% increase in the level of T947 phosphorylation in wild-type, but only a small increase (10%) in the *tmk1/4* mutant. Thus, TMK1 and TMK4 are the major regulators of auxin-induced AHA phosphorylation and activation. However, we cannot rule out the possibility that TMK2 and TMK3 may have an overlapping function with TMK1/4 (Xu, T et al., 2014, Sciences, Dai, N et al., 2013, PLoS One), especially under certain environmental conditions or at specific developmental backgrounds, and may work together with the TIR1/AFB-SAUR pathway, contributing to AHA phosphorylation and activation when TMK1 and TMK4 are missing. In our future experiments, we will more vigorously test whether TMKs are indeed required for auxin-triggered initial phosphorylation of AHAs, while the TIR1/AFB-SAUR pathway sustains the AHA phosphorylation to a high level for an extended time period, but this is out of the scope of this current work.

Figure 2b should include statistical analysis.

Response: Statistical analysis is now included for Fig. 2b.

*Figure 4 presents evidence using a fluorescent pH indicator that *tmk1 tmk4* mutants have higher apoplastic (extracellular space) pH, consistent with the effects on proton pump phosphorylation. I find that Figure 3 does not add much that is not also in Figure 4, so these two figures could be combined, and all or part of the data in Figure 3 put in a supplementary figure. The pH changes with and without auxin in wild type and mutant are the most important aspect here, and the shorter hypocotyl and shorter cells of the *tmk1 tmk4* mutant have been reported before anyway.*

Response: We thank the referee for the suggestion to combine Figure 3 with Figure 4. This suggestion certainly makes a lot of sense, as both figures contain data related to the effect of the *tmk1/4* mutations on apoplastic pH and cell elongation. Key results in Fig. 3 are that TMK1/4 regulate apoplastic pH and that there is a correlation of changes in pH with those in cell lengths between wild type and *tmk1/4* mutants, whereas Fig. 4 focuses on roles of TMK1/4 in auxin-mediated apoplastic acidification. Furthermore, if we were to combine the two figures, we would also need to combine Ext data Fig 3 and 4. This would make this revised extended data figure too complicated. Because the number of main figures is within the Nature figure limit and because we feel that the current figure arrangement is easier for the reader, we would prefer not to change this arrangement. However, if the referee feels strongly about his/her suggestion to change the rearrangement, we would be happy to do it.

*Fig. 4c. It would be interesting to test growth stimulation of *tmk1 tmk4* mutant hypocotyls in response to auxin in whole seedling assays, not just the short-term assay with excised hypocotyl segments, which only gives a response of about 10%. Whole-seedling assays work best in light-grown seedlings using a synthetic auxin such as picloram.*

Response: We thank the referee for this excellent suggestion and have included this data in Fig. 4d.

Cao et al. reported previously that auxin induces cleavage of TMK1 to affect nuclear auxin responses. It would seem relevant to discuss how the cleavage might affect the H⁺-ATPase activation. Does the cleavage activate the portion of the kinase that is still at the PM, or perhaps inactivate it to then limit H⁺-ATPase activation?

Response: *Whether the cleaved kinase is active toward ATPase phosphorylation is an interesting question that is worthy of future investigation. We found that auxin induced an extremely rapid but stable TMK1-AHA1 interaction on the PM, which was determined by the protoplast-based FRET assay, indicating that the cleaved form of TMK1 is unlikely responsible for the activation of PM AHAs since the cleaved kinase domain is rapidly translocated to the nucleus.*

lines 147-150. Why would expression in E. coli cause phosphorylation of Thr947? If anything I would have expected the immunoprecipitated H⁺-ATPase to be phosphorylated already. Perhaps the E. coli-expressed protein folds incorrectly.

Response: This is a good question, but we did not attempt to pinpoint the reason for our detection of *E. coli* expressed AHA-C proteins by the pT947 antibody, which prevented us from assaying TMK's activity to phosphorylate AHA1-C *in vitro*. The referee correctly pointed out that we cannot rule out the possibility that the recombinant protein folds incorrectly, causing the non-specific detection by the pT947 antibody. To avoid the unnecessary and potentially confusing statement, we deleted this statement that was intended to rationalize our use of protoplast-expressed AHA1-GFP and synthetic AHA1-C peptide for testing the direct phosphorylation of AHAs by TMK1.

They present the apoplast pH data as relative fluorescence at two different wavelengths. Can the degree of pH change be estimated, for example by calibrating the dye with buffers?

Response: Indeed, the degree of pH change can be estimated by calibrating HPTS dye with buffers as suggested by this referee and carried out in the root system by Barbez et al (2016. PNAS). We also verified this by determining the ratiometric change of 458/405 according to different pH 1/2 MS buffer in hypocotyls with similar findings. We prefer to use the relative values in our manuscript because other quantitative data (such as phosphorylation and ATPase hydrolysis) are presented as relative values, allowing meaningful comparisons among different data sets between wild type and the *tmk* mutants. Furthermore, we are concerned about the possible confusing or misleading information if our pH data were presented as calibrated absolute pH values for the following reasons. First, we found, even in wild-type the absolute mean values of 458/405 intensity varied among

different sets of samples, likely due to slight differences in growth conditions and growth stages. Furthermore, the phenotypes of *tmk1 tmk4* double mutants are highly variable when observed among different lab conditions (e.g., double mutant seeds were very difficult to germinate in Jiri Friml's lab, but this did not happen in our lab). Thus it is quite likely the calibrated absolute pH values for the mutant greatly vary among different conditions. Therefore the use of the relative pH values used in our work is sufficient for the interpretation and conclusion of our results and to avoid potential confusion in the literature.

Referee #3 (Remarks to the Author):

Auxin mediated cell expansion has been the subject of intense research for decades and the acid growth hypothesis has been central to helping explain the rapid (i.e. minutes) non-genomic stimulus of growth. However, the molecular mechanism(s) regulating auxin mediated cell expansion remains poorly understood.

The current manuscript by Lin et al throws new light on this previously intractable problem through their characterization of the PM-localized TMK receptor-like kinases (and its interactive partners/targets). Using an IP-MS based approach, Lin et al show that TMK receptor kinases interact with members of AHA gene family encoding PM ATPases. They further show that TMK kinase domain (but not the extracellular domain) interacts with AHAs in an auxin dependent fashion using an elegant FRET based approach. It is known that the phosphorylation of the conserved penultimate threonine (Thr) residue in the AHA family members is crucial for the activation of these PM-ATPases. Lin et al show that auxin activates AHA phosphorylation through TMKs.

*Phosphorylation of the penultimate Thr is known to be crucial for the activation of AHAs. Pull down assays using anti-pThr-947 antibody recognizing penultimate Thr, Lin et al show that the level of phosphorylated penultimate Thr was reduced in the *tmk1/4* mutant. Moreover, the activation of PM H⁺-ATPase is coupled with ATP hydrolysis and this was abolished in the *tmk1/4* mutant. Uptake of lithium ions is coupled with the activation of PM ATPase and in a very elegant experiment, Lin et al show that *tmk1/4* mutant seedlings were more tolerant to toxic lithium ions than the wild type. The authors then show that auxin mediated apoplast acidification is correlated with increase in hypocotyl cell length. This auxin mediated apoplastic acidification and increase in hypocotyl cell length is abolished in *tmk1/4* mutants but is restored when complemented with TMK1. This provides a direct mechanism for acidification of apoplast and cell expansion.*

The research is of high quality and is of general interest to plant biologists. The manuscript is well-written, and experiments are logical and well designed. There are some issues with the manuscript that need to be resolved to further improve the quality of the manuscript.

We thank the referee for the positive comments on our work.

One of the key concerns is that the auxin dependent nature of AHA-TMK interaction is performed in protoplasts. Given that protoplasts don't have cell walls and the nature of this experiment in interpreting this is as an auxin dependent acid-cell response, should this not be done in a system with a cell wall? How realistic are the conditions on the plasma membrane of a protoplast? What type of cells are these? Have they de-differentiated?

Response: We thank the referee for these interesting and excellent questions regarding the use of protoplasts in our co-IP assays. We were interested in the most rapid response of this interaction to auxin treatments, and protoplasts provide an ideal system for this assay. In contrast to tissues with cuticle/wax surface that prevent auxin from accessing the cell surface, auxin most rapidly access to the outer surface of the plasma membrane in protoplasts. Furthermore, protoplasts can be instantly frozen in liquid N₂ any time after auxin addition. Finally, in combination with microfluidic devices, protoplasts provide a necessary tool for analyzing rapid and dynamic responses involving single cell tracking, as shown in our new FRET analysis (Fig. 1e-g). Therefore, the protoplasts system can provide information on the initial/rapid responses of TMKs/AHAs to auxin. However, as the referee pointed out, it is possible that the protoplast system does not fully reflect the intact tissue or whole plant system because the cell wall or the whole plant system may feedback regulate the TMK-AHA interaction that is not present in the protoplast system. Whether there is a feedback regulation involving the cell wall (or cell wall mechanics) will be an interesting question for future investigation.

The protoplasts we used were prepared from the leaves of 4-week-old soil-grown plants, and were used within 8 hours after they were generated. Thus, it is unlikely that they are de-differentiated.

*Despite this concern, the experiments presented in figures 3 and 4 where the authors show that auxin mediated apoplast acidification is correlated with increase in hypocotyl cell length and this auxin mediated acidification of apoplast and increase in hypocotyl cell length is abolished in *tmk1/4* mutants but rescues by *TMK1* is convincing and provides a direct mechanism for apoplastic acidification and cell expansion.*

We thank the referee for this positive comment.

*It is clear that penultimate Thr is crucial for the function of AHAs and authors have shown that pull down assays using anti-pThr-947 antibody that the level of phosphorylated penultimate Thr was reduced in the *tmk1/4* mutant. It would be interesting to check the hypocotyl cell elongation upon auxin treatment in *tmk1/4* mutants complemented with kinase dead *TMK1*.*

Response: We thank the referee for the constructive suggestion. We attempted to obtain *TMK1km tmk1-1 tmk4-1* line by crossing *TMK1km tmk1-1* to *tmk1-1 tmk4-1* mutant, but have failed to obtain this line to date. We will continue to obtain this material for testing whether kinase activity is required for TMK modulated auxin-induced hypocotyl elongation in the future. Given the phosphorylation of AHAs by the kinase activity of TMKs, it is unlikely that AHA activation is regulated independently of the kinase domain.

The phosphorylation experiments presented in figure 2 are generally well described but it is not clear which tissues have been used for these assays. Though this information is present in the figure legend, it will be useful to mention this in the main text too.

Response: We thank the referee for the suggestion. The relevant information has been added in the main text (lines 148-149).

In some of the experiments, the authors have used non physiological concentration of NAA (fig1 1uM NAA). This is not ideal and it will be useful to repeat the assays using more physiological concentrations.

Response: We appreciate this constructive suggestion. We showed that auxin is able to induce an increase in the level of phosphorylated T947 in wild type seedlings at both low (100 nM Ext Data Fig. 2d,e) and high (10 μ M, Fig 2. a, e) concentrations in a TMK1/4-dependent manner. Low auxin concentrations (100 nM) also rapidly induced interaction between TMKs and AHAs (Fig. 1). These results suggest that TMKs are regulated by physiological levels of auxin and are required for auxin-induced PM-H⁺-ATPase activation at different concentrations.

Pull down assays using pTMK1::TMK1-GFP lines are very nice (fig1 A) but am puzzled why the authors decided to use 35S::TMK1:GFP plants (Fig 1B) when they also had the opportunity to use pTMK1::TMK1-GFP lines that provides more spatial resolution. Also, whilst I understand the authors were interested in AHAs, there is value in also reporting what other proteins came out in these TMK immunoprecipitation assay.

Response: We did not use *35S::TMK1:GFP* plants in the co-IP experiment shown in Fig. 2b. Instead we used *35S::GFP-AHA1* in this experiment, and determined the presence of native TMK1 in the IP complex by using anti-TMK1 antibody. We have the *35S::GFP-AHA1* transgenic line in hand and did not have a *pAHA1::GFP-AHA1* transgenic line available. It is noteworthy that AHA1 and AHA2 are widely expressed in all tissues, as is the activity of the 35S promoter. Our co-IP results with the *35S::GFP-AHA1* line should reflect the native interaction between AHA1 and TMK1. Importantly, the reciprocal co-IP experiments were carried out using the *pTMK1::TMK1-GFP tmk1-1 tmk 4-1* line (Ext data Fig. 1b). Therefore,

altogether our results demonstrate an interaction between AHA1 and TMK1 under the native physiological conditions.

As discussed in responses to referee #2 above, we prefer not to include other proteins present in the immunoprecipitates that have not been confirmed by independent methods due to possible false positives in order to avoid misleading information. Nonetheless we would be happy to provide them if this referee feels strongly about it.

The experiments with toxic lithium ions (extended figure 3) and fusaric acid are very elegant. I suggest authors consider moving the Lithium data from extended figure 2 to the main figure 2.

Response: We thank the referee for appreciating this experiment. Given the view of referee #1 (see major point #2), we hope that this referee is fine with leaving this data in Ext data Fig. 2.

The authors also need to provide more details in the methods section. For example, it is not clear which tissues have been used for Co-IP studies (eg whole plants, leaves??). Also, there are no methods provided for FRET analysis. Information on what statistical tests have been done is often vague and it would be useful that this information is provided in the figure legends.

Response: We thank the referee for the suggestion. We added more detailed information in the Materials and Methods (see line #418-419,509, and 574-602), including FRET analysis, and statistical test methods were provided in the figure legends.

Reviewer Reports on the First Revision:

Referee #1 (Remarks to the Author):

The authors have addressed all concerns that I have raised. I commend them for their excellent work.

Referee #2 (Remarks to the Author):

The work shows that the Arabidopsis TMK1 receptor kinase can phosphorylate AHA H⁺-ATPases at a key activating regulatory site, and that TMK levels correlate with in vivo proton pump phosphorylation level, activity, and cell growth. This is an important conclusion that will likely become central to our thinking of growth control in plants. The authors have added some new data to strengthen the previous version, most notably a FRET assay using microfluidics that allows

detection of interaction between H⁺-ATPase and TMK within 1 minute of auxin application. It would be interesting also to use this system to test whether washout of auxin reverses the interaction.

Although two reviewers suggested it, they have not added a table showing the full list of proteins detected in the immunoprecipitation screen. They cite a desire to avoid introducing unvalidated results into the literature, but this seems a disingenuous argument that just suppresses information. We could gain perspective on what proportion of hits were H⁺-ATPases, and also learn (by comparison to other datasets) what other proteins may be potential common artifacts. By analogy, one would not publish an RNA-Seq dataset without making the full results available, even if only certain genes were pursued.

line 150 - they list several AHA proteins whose phosphorylation is affected, but the corresponding figure (S2b) shows that only three of these have statistically significant differences from wild type. So, the text should be adjusted to reflect that.

I noticed typos in lines 295 and 320.

Referee #3 (Remarks to the Author):

The manuscript reports highly original and significant results which will literally 'rewrite the text books' when it comes to the field's understanding of auxin action - specifically with respect to rapid, non-genomic mechanisms - which have remained unresolved until now.

The authors have also responded positively and constructively to all of the 3 reviewers comments, prompting the addition of new data, revisions to the text or comprehensive rebuttal of selected points, delivering a much improved manuscript.

I recommend acceptance of the revised manuscript given the significance of the results reported and improvements to the current manuscript.

Author Rebuttals to First Revision:

Summary of responses: We thank all three referees for your overall positive comments on our work and your constructive critiques, which have certainly helped to improve our manuscript. Please note that we have addressed the concerns that require additional information and changes. Please see below for our point-by-point responses in blue.

Referees' comments:

Referee #1 (Remarks to the Author):

The authors have addressed all concerns that I have raised. I commend them for their excellent work.

Response: We appreciate this referee for the positive comment on the significance of our work.

Referee #2 (Remarks to the Author):

The work shows that the Arabidopsis TMK1 receptor kinase can phosphorylate AHA H⁺-ATPases at a key activating regulatory site, and that TMK levels correlate with in vivo proton pump phosphorylation level, activity, and cell growth. This is an important conclusion that will likely become central to our thinking of growth control in plants. The authors have added some new data to strengthen the previous version, most notably a FRET assay using microfluidics that allows detection of interaction between H⁺-ATPase and TMK within 1 minute of auxin application. It would be interesting also to use this system to test whether washout of auxin reverses the interaction.

Response: We thank this referee for the positive comment on the significance of our work.

We appreciate the referee's constructive suggestion to add wash out step on auxin-induced TMK-AHA rapid interaction determined by FRET assay. To achieve that, a new microfluidic device with an additional inlet needs to be designed and produced, which will take months. Importantly we have already including a mock control to show that the effect of auxin application on the interaction is specific. Given the interest of publishing this work timely, I hope that you would agree that we do not need to conduct the washout experiment at this revision. We will continue to produce the new-designed microfluidics for testing whether washout of auxin can reverse the interaction between TMK and AHA on the PM in the future.

Although two reviewers suggested it, they have not added a table showing the full list of proteins detected in the immunoprecipitation screen. They cite a desire to avoid introducing unvalidated results into the literature, but this seems a disingenuous argument that just suppresses information. We could gain perspective on what proportion of hits were H⁺-ATPases, and also learn (by comparison to other datasets) what other proteins may be potential common artifacts. By analogy, one would not publish an RNA-Seq dataset without making the full results available, even if only certain genes were pursued.

Response: We agree with this reference that the IP-MS data may be of interest to the readers. A full list of TMK1-GFP IP-mass spectrometry identified candidate interactors was prepared and submitted as Supplementary Information Table.2.

line 150 - they list several AHA proteins whose phosphorylation is affected, but the corresponding figure (S2b) shows that only three of these have statistically significant differences from wild type. So, the text should be adjusted to reflect that.

Response: We appreciate the referee's suggestion to adjust the description. We have now listed only the statistically significant AHAs in the text to reflect the finding (line 109).

I noticed typos in lines 295 and 320.

Response: We thank the referee for pointing out typos. The relevant typos have been corrected in the main text (lines 210, and 219)

Referee #3 (Remarks to the Author):

The manuscript reports highly original and significant results which will literally 'rewrite the text books' when it comes to the field's understanding of auxin action - specifically with respect to rapid, non-genomic mechanisms - which have remained unresolved until now.

The authors have also responded positively and constructively to all of the 3 reviewers comments, prompting the addition of new data, revisions to the text or comprehensive rebuttal of selected points, delivering a much improved manuscript.

I recommend acceptance of the revised manuscript given the significance of the results reported and improvements to the current manuscript.

Response: We thank the referee for the positive comments and the recommendation on our work.